# Directing curli polymerization with DNA origami nucleators

Xiuhai Mao [1,2,6], Ke Li[1], Mengmeng Liu[3], Xinyu Wang[1], Tianxin Zhao[1], Bolin An[1], Mengkui Cui[1], Yingfeng Li[1], Jiahua Pu[1], Jiang Li[2], Lihua Wang[2,3], Timothy K. Lu[4], Chunhai Fan [5] & Chao Zhong [1]

The physiological or pathological formation of fibrils often relies on molecular-scale nucleators that finely control the kinetics and structural features. However, mechanistic understanding of how protein nucleators mediate fibril formation in cells remains elusive. Here, we develop a CsgB-decorated DNA origami (CB-origami) to mimic protein nucleators in *Escherichia coli* biofilm that direct curli polymerization. We show that CB-origami directs curli subunit CsgA monomers to form oligomers and then accelerates fibril formation by increasing the proliferation rate of primary pathways. Fibrils grow either out from (departure mode) or towards the nucleators (arrival mode), implying two distinct roles of CsgB: as nucleation sites and as trap sites to capture growing nanofibrils in vicinity. Curli polymerization follows typical stop-and-go dynamics but exhibits a higher instantaneous elongation rate compared with independent fibril growth. This origami nucleator thus provides an in vitro platform for mechanistically probing molecular nucleation and controlling directional fibril polymerization for bionanotechnology.

[1] Materials and Physical Biology Division, School of Physical Science and Technology, ShanghaiTech University, Shanghai 201210, China. [2] Division of Physical Biology and Bioimaging Center, Shanghai Synchrotron Radiation Facility, Shanghai Institute of Applied Physics, Chinese Academy of Sciences, Shanghai 201800, China. [3] Shanghai Key Laboratory of Green Chemistry and Chemical Processes, School of Chemistry and Molecular Engineering, East China Normal University, 500 Dongchuan Road, Shanghai 200241, China. [4] Synthetic Biology Group, Research Laboratory of Electronics, Massachusetts Institute of Technology, Cambridge, MA 02139, USA. [5] School of Chemistry and Chemical Engineering, and Institute of Molecular Medicine, Renji Hospital, School of Medicine, Shanghai Jiao Tong University, Shanghai 200240, China. [6] Present address: Institute of Molecular Medicine, Renji Hospital, School of Medicine, Shanghai Jiao Tong University, Shanghai 200127, China. Correspondence and requests for materials should be addressed to C.F. (email: fanchunhai@sjtu.edu. cn) or to C.Z. (email: zhongchao@shanghaitech.edu.cn)

Molecular nucleators are often harnessed by living cells in the construction of filament structures or fibrous networks with controlled nucleation and aggregation kinetics, leading to appropriate structural features and physiological functions[1–6]. Besides their critical roles in biological functions, protein nucleators also have broad and important implications in pathology, disease, and bionanotechnology[7,8]. One example is the outer membrane-localized amyloidogenic curli-specific gene B (CsgB) proteins, which serve as nucleators guiding the polymerization of the major protein subunit CsgA into curli fibrous networks on the cell surface, essentially contributing to biofilm formation in *Escherichia coli*[9–11]. Note that in the absence of CsgB, CsgA is secreted from the cell as monomers, and no fibrils are formed adjacent to cells[12]. As a product of a highly regulated and directed process, curli represent a twist to the conventional view of amyloidogenesis, in which disease-associated amyloid formation is considered a protein misfolding event[13–15]. Another example is actin-related protein complex (ARP2/3), a handful of nucleation-promoting factors and formins, which precisely controls the formation of the actin cytoskeleton in living cells[16]. The spatially directed assembly of actin filaments or amyloid proteins via controlled nucleation processes has been applied to the fabrication of three-dimensional electrical connections and nanostructures in which amyloid fibrils were sheathed within DNA origami nanotubes[17,18].

Although remarkable advances have been made in structural and functional analyses of several types of molecular nucleators[17], a deeper mechanistic understanding of how protein nucleators regulate fibril assembly on the single molecule level remains elusive. Previous studies have suggested a nucleation role of CsgB in *E. coli* biofilm formation based on both in vitro and in vivo experiments (Fig. 1a and Supplementary Figure 1)[10,12,19,20]. However, as they are based on ensemble measurements for probing nucleation and growth mechanisms using absorption or fluorescence spectroscopy, those methods often lack detailed molecule-scale information on the primary nucleation steps. In addition, the recording of CsgB-directed curli fibril polymerization unavoidably interferes with both CsgA and CsgB self-assembly in such ensemble measurements. By tracking individual fibril growth based on high-speed atomic force microscopy (HS-AFM), Sleutel et al.[21] recently revealed important growth characteristics and kinetics of curli polymerization. However, such real-time in situ nanoscopic imaging alone would not differentiate nucleator-mediated fibril growth from independent fibril growth and thus cannot be directly applied to probing nucleator-directed nucleation and growth processes. Indeed, their work did not investigate curli polymerization in the presence of the nucleator protein CsgB. Molecular-scale understanding of the specific roles of CsgB in curli formation therefore remains a daunting challenge.

The state-of-the-art DNA origami technique provides a versatile platform for studying molecular interactions[22,23]. Its prominent features include programmable shapes and geometries, as well as site-directed decoration with various functional entities[24–28]. Recent advances include the use of DNA origami frames for targeted genetic phasing of long-range haplotypes, the application of DNA origami for probing changes within chromatin at the single-molecule level, and the construction of self-assembling DNA nanotubes to connect molecular landmarks[29–32].

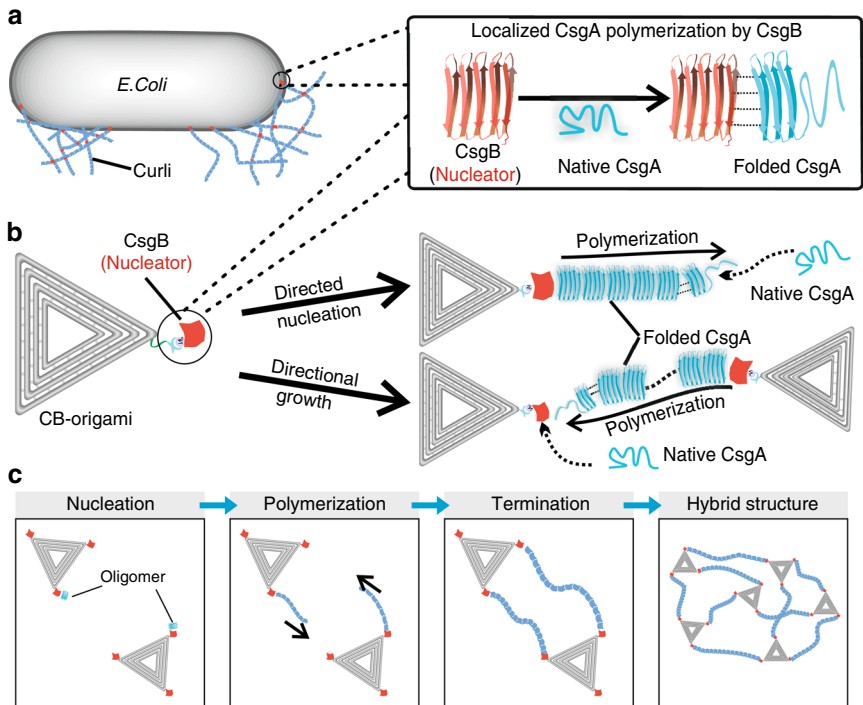

**Fig. 1** Directional polymerization of CsgA with DNA origami molecular nucleators. **a** Schematic of curli biogenesis associated with *Escherichia coli* biofilm formation. Curli are extracellular proteinaceous functional amyloid aggregates produced by certain enteric bacteria (*E. coli* and *Salmonella* spp). CsgA and CsgB are the major and minor protein subunits of curli fibers. CsgB, anchored on the cell surface by membrane proteins, serves as a nucleus, triggering CsgA polymerization. **b** Schematic of designer DNA origami as a molecular nucleator for probing the molecular nucleation of CsgA proteins. CsgB proteins are anchored at the vertex of triangular DNA origami through Metal-NTA-His coordination bonds. Directional fibril polymerization predominantly occurs in the presence of CB-origami, which not only accelerates the nucleation process by promoting oligomer formation but also guides the directional growth of a proliferating fibril during fibril elongation. **c** Schematic showing the rational design of DNA origami/fibril complex network structures through directional polymerization of CsgA monomers using designer CB-origami as the template

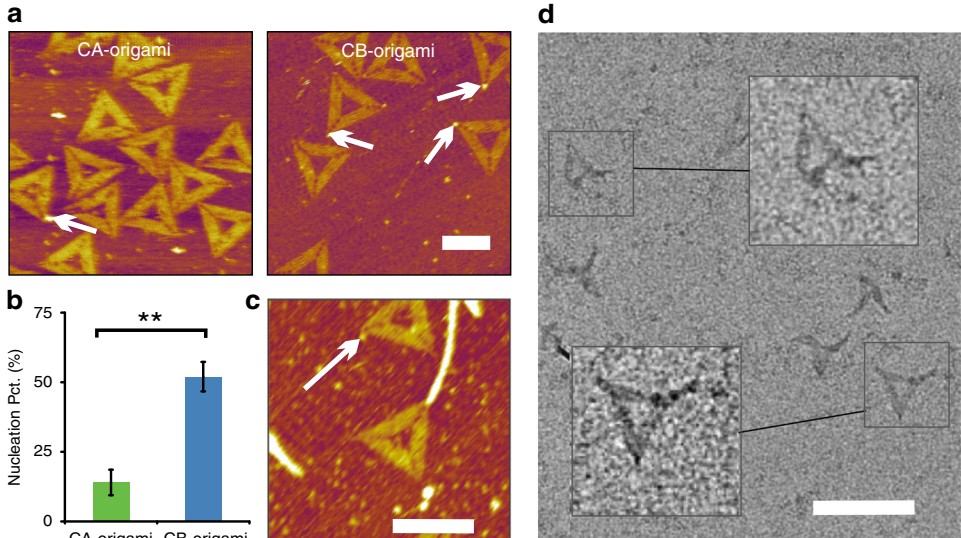

**Fig. 2** Probing the nucleation roles of CsgB in CsgA polymerization with CB-origami. **a** AFM height image of the formed oligomers (3.2 ± 0.3 nm in height) at the vertex of CA-origami and CB-origami, respectively (with 1.0 μM CsgA monomers). **b** Comparison of the nucleation efficiency between CB-origami (52.6 ± 5.3%) and CA-origami (14.3 ± 4.6%) by calculating the percentage of origami tethered with dot-like oligomer structures at the vertex (with 1.0 μM CsgA monomers); the data were collected by counting 100 origami structures for each group. All data points were averages of three independent experiments (n = 3) and presented as the mean ± standard error of the mean (s.e.m.). **P < 0.01, two-tailed two-sample t-test. Source data are provided as a source data file. **c**, **d** AFM height and TEM images showing fibril (3.4 ± 0.4 nm in height) growing directly from the vertex of the CB-origami (with 2.0 μM CsgA monomers). The white arrows refer to the dot-like oligomers formed at the vertex. Scale bars: 100 nm. Note: AFM imaging was purposely carried out in tapping mode in liquid

Here, using *E. coli* curli amyloid as a model molecular nucleator system, we probe CsgB-directed curli polymerization at single-fibril resolution by coupling designer DNA origami-based molecular nucleators with in situ high-speed AFM imaging. The molecular nucleators are designed by anchoring CsgB proteins onto triangular DNA origami in a position-defined manner (Fig. 1). We hypothesize that the DNA-origami-anchored CsgB would promote the local nucleation rate of CsgA by lowering the energy barrier for β-strand formation of the free CsgA monomers in the vicinity of CsgB, thereby increasing the overall rate of nanofibril polymerization. The resulting directional curli polymerization enabled by CsgB-decorated DNA origami (CB-origami) closely recapitulates molecular interactions between CsgA and CsgB in *E. coli* biofilms (Fig. 1a). Importantly, the triangular DNA origami also serves as a molecular landmark to facilitate localization of the tethered CsgB, rendering feasible the accurate capturing and tracking of individual nucleator proteins and subsequent fibril elongation (Fig. 1b). We demonstrate that, by coupling the designer DNA origami technique with real-time in situ nanoscopic imaging, we can accurately track molecular nucleator-directed polymerization of CsgA at single-fibril resolution. In addition, we construct network structures of DNA origami/amyloid fibril complexes by site-specifically inducing curli polymerization using designer CB-origami as organic templates (Fig. 1c).

## Results

**CB-origami nucleators direct curli polymerization.** We designed triangular DNA origami (edge length: 120 nm and height: 1.5 nm), with prescribed DNA link strands at the vertex. The DNA link strands were then hybridized with DNA capture strands tagged with a functional group containing nitrilotriacetic acid (NTA), resulting in NTA-decorated DNA origami (Supplementary Figure 6-7)[33]. This functionalized DNA origami could further interact with the CsgB protein (with poly-histidine tags

appended at the C-terminus) to form CsgB-decorated DNA origami (CB-origami) through Metal-NTA-His coordination bonds. CsgA-His and CsgB-His molecules (referred to henceforth as CsgA and CsgB, respectively), applied in this study were in monomer form and had the expected molecular weights, as confirmed with SDS-PAGE (sodium dodecyl sulfate polyacrylamide gel electrophoresis) and Western blotting (Supplementary Figure 8). In particular, we applied formic acid (FA) to dissolve and store CsgA monomers instead of the widely used guanidine hydrochloride (GdHCl) in our study, as, unlike GdHCl-stored CsgA, the FA-stored CsgA does not require an additional time-consuming desalting process. Consistent with several previous studies[34,35], our His-Tag fused CsgA proteins apparently had normal self-assembling capacity to form amyloid nanofibrils with typical cross-β structures and resembled the typical fibril structures present in wild-type biofilms (Supplementary Figure 9).

To assess the nucleation role of CsgB in directing CsgA polymerization, we probed the morphology of CsgA proteins in proximity to CB-origami with AFM by incubating CsgA monomers with varied concentrations (1.0–5.0 μM) together with CB-origami. The number of independently formed CsgA aggregates significantly increased with the increase of CsgA concentration due to the concentration dominant kinetic effect during CsgA polymerization (Supplementary Figure 11), in agreement with the kinetic model that we constructed based on data acquired from thioflavin T (ThT) fluorescence assay (Supplementary Figure 12). To better assess the nucleation events of CsgB, we therefore imaged formed CsgA aggregates together with CB-origami in lower concentration of CsgA, such as 1.0 μM and 2.0 μM, which apparently is in its nucleation phase. Indeed, we observed dot-like aggregates, with an average height of 3.1 ± 0.2 nm, formed at the vertex of CB-origami in 1.0 μM CsgA monomer solution (Fig. 2a, Supplementary Figure 13 and 14, and Supplementary Movie 1), while CsgA nanofibrils, with an average height of 3.4 ± 0.4 nm, appeared to be growing directly out from

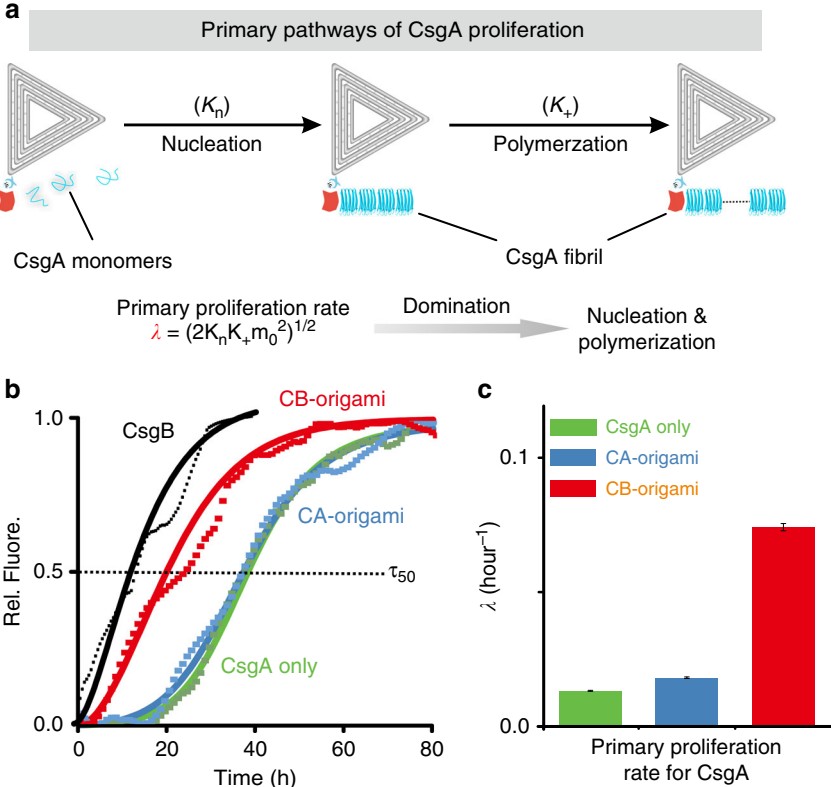

**Fig. 3** Kinetic analysis of CsgA polymerization with and without CB-origami. **a** Schematic showing the primary pathways of CsgA polymerization in the presence of CB-origami and associated rate constants for the nucleation ($K_n$), elongation ($K_+$), and combined proliferation processes ($\lambda$). Note: $\lambda$ is related to the rate of formation of new aggregates through primary pathways, $m_0$ is the initial concentration of soluble monomers. **b** ThT assay revealing the polymerization kinetics of CsgA monomers (5.0 µM) for three parallel but independent samples: CsgA alone (light green), CsgA added with CB-origami (red), CsgA added with CsgB (black), or CsgA added with CA-origami (light blue). Both CB-origami and CsgB increase the kinetic rate of CsgA polymerization. The data (dotted curves), representative of the cumulative normalized fluorescence signal of the samples, were fitted with a sigmoidal function (solid-line curves) using the supplementary equation 1 in the supplementary method. The black dash line represents $\tau 50$, the time for aggregate mass to reach 50% of its final weight. **c** Comparison of the kinetic rates controlling proliferation of CsgA through primary pathways for CsgA only (light green), CsgA added with CA-origami (light blue) and CsgA added with CB-origami (red), and presented as the mean ± standard error of the mean (s.e.m) based on three independent experiments ($n = 3$). Source data are provided as a Source Data file

the vertex of CB-origami when the concentration of CsgA increased above 2.0 µM (Fig. 2c, d). As both dot-like aggregates and fibril structures were found to form at the vertex of CB-origami at 2.0 µM CsgA (Fig. 2c and Supplementary Figure 13–16) and have identical height, which is significantly higher than that of origami (1.5 nm), we reasoned that they were products of CsgA in different stages during polymerization. Conceivably, the dot-like aggregates had been oligomeric structures at an early stage, while fibrils were formed as final products by the continuous attachment of CsgA monomers to the growing ends of the intermediate oligomer structures. In addition, when incubated in solution containing NTA-decorated Au nanoparticles (AuNPs), fibrils did bind the AuNPs apparently because of the histidine tags, suggesting that the formed fibrils were indeed assembled from CsgA monomers (Supplementary Figs. 17 and 18)[36].

As oligomer is necessarily an early-stage product in CsgA polymerization and directly associated with the specific nucleation role of CB-origami, we assessed the nucleation efficiency of CB-origami by calculating the percentage of origami structures tethered to oligomers at the vertex of CB-origami. Previous studies have shown that, relative to the influence of CsgB, the presence of free CsgA does not exhibit a significant influence on overall CsgA polymerization[12,19], so we here used CsgA-decorated DNA origami (CA-origami) as a control under the

same condition (1.0 µM CsgA solution). We found that the percentage of oligomer-tethered CB-origami reached 52.6 ± 5.3%, significantly higher than that of CA-origami, with a percentage of 14.3 ± 4.6% (Fig. 2b and Supplementary Figure 19). This result thus indicated a higher nucleation efficiency of CB-origami compared with CA-origami, confirming the role of CB-origami as the predominant molecular nucleator for CsgA polymerization.

To further elucidate how CB-origami affected the nucleation kinetics of CsgA, we performed a thioflavin T (ThT) assay, a fluorescence dye assay that is often used to monitor the kinetics of formation of amyloid structures from soluble monomers (Fig. 3b)[37]. The data revealed that the addition of CB-origami led to the shifting of the ThT curve to the left, while the addition of CA-origami caused no noticeable changes. In particular, $\tau 50$, the time for aggregate mass to reach 50% of its final weight, decreased from 37.8 ± 1.5 to 21.9 ± 1.1 h in the presence of CB-origami, implying that CB-origami indeed significantly accelerated curli polymerization.

Polymerization of amyloid proteins is the consequence of a series of microscopic events, including primary nucleation, fibril elongation, and secondary nucleation processes, the rates of which are dependent on the rate constants $K_n$, $K_+$, and $K_2$, respectively[38,39]. These processes can be generally divided into two independent polymerization pathways: primary and secondary (Fig. 3a and Supplementary Figure 21). In primary

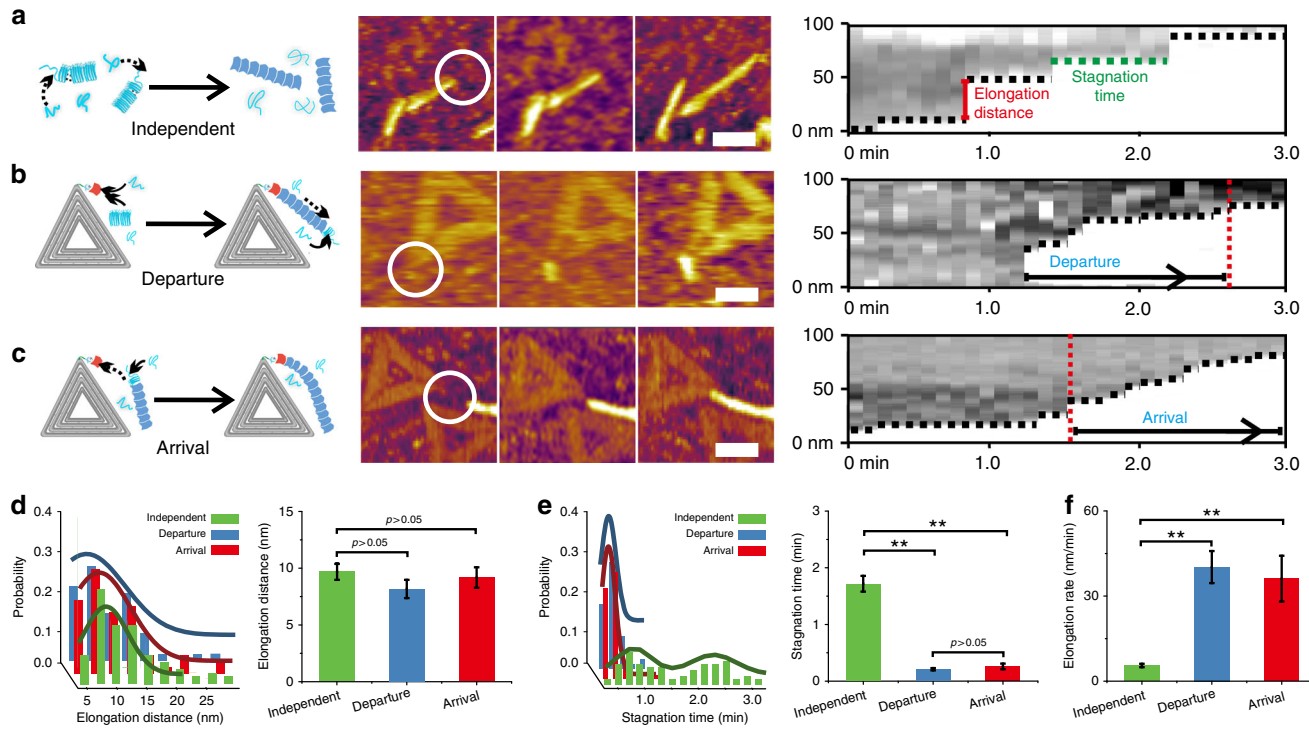

**Fig. 4** In situ probing of CsgA polymerization in the presence of CB-origami. **a–c** Schematic, representative AFM time-lapse images and corresponding kymograph images showing morphological evolution of CsgA fibril formation either **a** in the absence of CB-origami (independent mode) or **b**, **c** in the presence of CB-origami: for fibril polymerization with CsgA monomers alone, an independent growth mode was typically observed, in which fibril growth occurs at both ends and typically follows a stop-and-go dynamics characterized by **a** a relatively long stagnation time for each step, **b** a departure mode typically observed during fibril growth in the presence of CB-origami, in which oligomers initially form at the nucleation site of DNA origami, followed by fibril elongation from the nucleation site, and **c** an arrival mode typically observed during fibril growth in the presence of CB-origami, in which an approaching fibril tends to grow towards and eventually terminates at the nucleation site of CB-origami. Representative traces of fibril elongation as a function of time in the independent, departure and arrival mode are shown on the right. **d–f** Comparison of the **d** elongation distance, **e** stagnation time, and **f** elongation rate for fibril growth in the independent, departure and arrival mode. Data for elongation distance and stagnation time were presented as the mean ± s.e.m., collected by counting eight different fibers for independent mode (92 growth steps in total), 8 different fibers for departure mode (50 growth steps in total), and 9 different fibers for arrival mode (57 growth steps in total). These results were presented as the mean ± s.e.m. **$P < 0.01$, two-tailed two-sample $t$-test. Data for elongation rate were calculated by dividing the elongation distance by the stagnation time. Note: these captured images (about 6 s per frame) and the corresponding kymograph images were produced based on the AFM videos in tapping mode in liquid, associated with Supplementary movie 2–5. Scale bars: 50 nm. Source data are provided as a Source Data file

polymerization pathway, new aggregates form at a rate dependent on the concentration of monomers and independent of the concentration of existing fibrils. Specifically, the polymerization rate of primary pathways can be described by $\lambda = (2K_n\,K_+\,m_0{}^2)^{1/2}$, where $\lambda$ is a combined parameter revealing primary polymerization rate and $m_0$ is the initial concentration of soluble monomer (Fig. 3a)[40].

The addition of nucleators to CsgA monomers accelerates nucleation, thereby resulting in kinetic change in the polymerization pathways. To further quantitatively assess the nucleation effect of CB-origami on kinetics of curli polymerization, we analyzed all the ThT data and determined the kinetic parameters based upon an established global fitting approach (Fig. 3b and Supplementary Figure 22 and 23)[40,41]. The $\lambda$ value for CsgA polymerization in the presence of CB-origami increased by fivefold (from $1.32 \pm 0.02 \times 10^{-2}\,\mathrm{h}^{-1}$ to $7.42 \pm 0.14 \times 10^{-2}\,\mathrm{h}^{-1}$), in stark contrast with only a 35% increase ($1.81 \pm 0.02 \times 10^{-2}\,\mathrm{h}^{-1}$) in the presence of CA-origami, likely arising from a non-specific heterogeneous nucleation effect[42]. The data thus suggested that CB-origami indeed served as functional and specific molecular nucleators that could accelerate CsgA polymerization, rather than serving as an artefactual impurity type of nucleus as did CA-origami. Noticeably, the $\kappa$ value, a combined

parameter representing the proliferation rate through secondary pathways (Supplementary Figure 21), decreased with the addition of CB-origami or CA-origami in both cases apparently because extra nucleators had been introduced, consistent with previous findings that the dominant status of the secondary polymerization gradually vanished with the addition of a nucleus (Supplementary Figure 24)[43,44].

**In situ probing CB-origami-directed CsgA polymerization.** To probe the underlying mechanism of molecular nucleator-directed polymerization of CsgA at single-fibril resolution, we monitored the in situ polymerization of CsgA in the presence of CB-origami with HS-AFM[45,46]. The real-time in situ AFM imaging, with time-lapse images captured about every 6 s, could inspect how nanofibrils formed at the vertex of CB-origami at the single fibril level. Based on the collected time-lapse AFM images (similar results based on at least four videos for each sample), we concluded that the origami/fibril hybrid structure, as shown in Fig. 2b, formed from two different growth modes: the departure and arrival mode, respectively (Fig. 4 and Supplementary Figure 25). In the departure mode, CB-origami initially served as molecular nucleators to induce the formation of nascent CsgA aggregate structures that further triggered the polymerization of

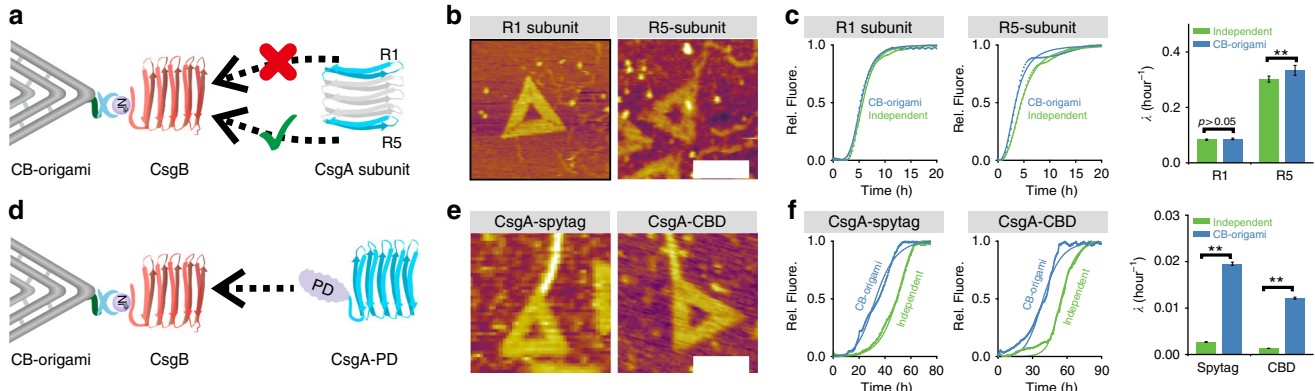

**Fig. 5** Directional polymerization of CsgA subunit and derivatives with CB-origami. **a–c** Nucleation-directed polymerization of CsgA subunits with CB-origami: **a** schematic showing CB-origami-directed polymerization of CsgA subunit, R1 and R5 domain, **b** Corresponding AFM height images of the assembled structures, **c** Representative ThT curves and kinetic rate constant comparison; **d–f** Nucleation-directed polymerization of functional CsgA proteins with CB origami: **d** schematic showing CB-origami-directed polymerization of CsgA-PD (PD, Peptide domain or Protein domain) proteins, CsgA-spytag and CsgA-CBD, **e** AFM height images of the assembled structures, showing that fusion domains did not block the nucleation effect of CsgB, **f** Representative ThT curves and kinetic rate constant comparison. Note: For all the ThT curves and kinetic constant comparison in **c** and **f**, green and blue curve/bar chart refers to independent polymerization and polymerization in the presence of CB-origami, respectively. $\lambda$ refers to the proliferation rate of primary pathways and presented as the mean ± s.e.m. based on three independent experiments ($n = 3$). **$P < 0.01$, two-tailed two-sample $t$-test. Scale bars: 100 nm. Note: AFM imaging was purposely carried out in tapping mode in liquid. Source data are provided as a Source Data file

CsgA, resulting in origami/fibril hybrid structure with elongated fibrils forming at the vertex of the origami (Fig. 4b, left and Supplementary Movie 2 and 3). In the arrival mode, when adjacent to the decorated CsgB, independent short fibrils elongated towards the vertex of CB-origami, eventually tethering to that vertex. In this mode, CB-origami served as terminus to guide the elongation direction of existing fibrils and even temporally retarded its elongation (Fig. 4c, left, Supplementary Figure 25 and Supplementary Movie 2-5). These two types of growth mode clearly implied that CsgB played two different roles in the CsgA aggregation processes: as nucleation sites and as trap sites that capture growing nanofibrils in their vicinity; both roles could accelerate the attachment of CsgA to the vertices of CB-origami.

To probe the variations in the instantaneous rate of fibril elongation over the course of the observation period, we constructed kymograph pseudo-images based on high-speed AFM imaging (Fig. 4a–c, right). A kymograph pseudo-image allows the dynamic process of a formed structure to be represented by extracting the pixel values along the trajectory of a final structure and linearizing the extracted points into columns that are stacked for the consecutive frames of a single movie[21]. We compared the kymographs representative of the two growth modes with the one based on independent fibril growth (without addition of CB-origami). Fibril growth in all cases exhibited stop-and-go dynamics characterized by periods of steady growth alternated with periods of stagnation. However, one clear difference was that fibril growth with CB-origami generally exhibited more frequent elongation steps compared with the fibrils growing independently (Fig. 4a–c). This observation became even more pronounced either when fibril growth initiated from the vertex in the departure mode or when a growing fibril approached the vertex in the arrival mode.

To further characterize the elongation behaviors, we measured both the elongation distances of each elongation step and the stagnation time between two growing steps, and calculated the instantaneous rate of elongation over the course of the observation period in these modes. Interestingly, fibril elongation in both growth modes followed irregular patterns characterized by variable distance in each step ranging from 2.0 nm to 25 nm, closely resembling that of independent fibril growth (Fig. 4d).

This observation thus suggested that the elongation distance in each step varied stochastically, independent of the presence of CB-origami. However, addition of CB-origami radically changed the average stagnation time (Fig. 4e). Specifically, comparison of the peak position and FWHM values for the distribution curves of both elongation distance and stagnation time in various modes (Table S1 and S2) revealed that the stagnation time for the departure and arrival modes are almost identical. In addition, there are two typical Gaussidan peaks for stagnation time in the independent mode of CsgA polymerization, consistent with the studies described by Sleutel et al. in which polymerization of isolated CsgA fibrils exhibits two unique growth speeds[21]. Although there are two peaks in the independent mode, the average stagnation time corresponding to each peak is significantly higher than that of the departure or arrival mode peaks, collectively suggesting that a relatively slower growth speed occurs in the independent growth mode.

In addition, we detected significant variations in the instantaneous rate of elongation: the mean rate of fibril elongation increased from $0.08 \pm 0.01 \, \text{nm s}^{-1}$ in the independent mode to $0.46 \pm 0.11 \, \text{nm s}^{-1}$ and to $0.54 \pm 0.09 \, \text{nm s}^{-1}$ in the departure and arrival modes, respectively (Fig. 4f). This elongation rate increase is a consequence of the fact that, as designed, the presence of the CsgB structures accelerates the formation of CsgA aggregates. We speculate that this acceleration results from the ability of CsgB to specifically interact with CsgA monomers in a manner that lowers their energy barrier for β-strand formation; this is reasonable as such a process would occur in the local vicinity of the DNA-origami nucleators and as such β-strands are known to promote polymerization[10,12]. Collectively, these results thus support that CB-origami could serve as an efficient molecular nucleator to significantly increase the proliferation rate in the primary pathway by accelerating the processes of CsgA monomers joining CB-origami and forming oligomers.

**Assessing nucleation of CsgA derivatives with CB-origami.** We next probed how CsgB interacted with CsgA by applying CB-origami to direct the polymerization of CsgA subunit domains (Fig. 5a). CsgA contains five imperfect repeating units (R1-R5)[47].

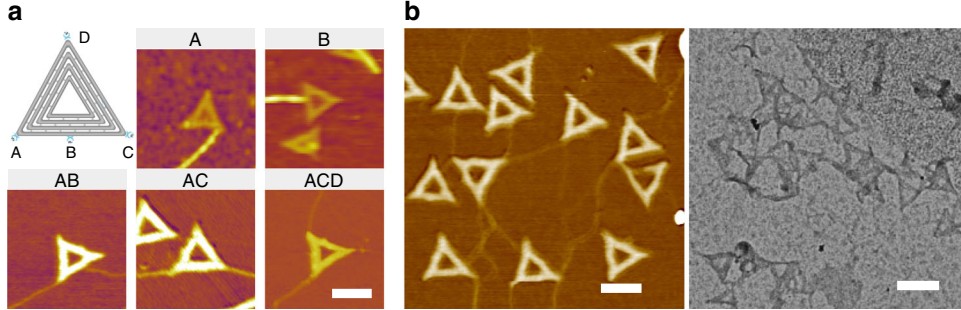

**Fig. 6** Nucleation-directed polymerization of CsgA with designer CB-origami. The assembled structures shown in AFM height and TEM images correspond to the schematics representative of the CB-origami structures applied. **a** Schematic of the applied origami structure containing a variable number of CsgB site-specifically anchored along the edge of the (left) triangular DNA origami and corresponding AFM images of the assembled structures. The annotated A, B, AB, AC, or ACD on top of the AFM images stands for the applied origami structure with CsgB anchored at the specifically designated positions shown in the schematic. **b** Typical DNA origami-fibril complex structures formed by polymerization of CsgA monomers (2.0 μM) in the presence of CB-origami, with CsgB tethered at the three vertexes. Scale bars: 100 nm. Note: AFM imaging was performed in tapping mode in air

The five stacked strand-loop-strand motifs form the amyloid core of CsgA. Previous studies suggested that either the R1 or R5 unit (marked in light blue) in CsgA responds to CsgB-mediated nucleation, and therefore directs CsgA to assemble on CsgB so that the resulting biofilm is closely associated with the cell surface of *E. coli*. However, it remains unknown whether it is the R1 or R5 subunit domain that directly interacts with CsgB and therefore is responsible for the CsgB-directed CsgA polymerization.

As chemically synthesized peptides bearing R1 and R5 sequences both can assemble into amyloid-like fibers in vitro, we directly tested and compared how CB-origami affected the in vitro polymerization of both subunit domains using ThT assays and AFM imaging. When R1 and R5 subunits were independently incubated with CB-origami and probed with AFM, only R5 fibrils (average height of 1.54 ± 0.14 nm) were found tethered to the origami at the designated positions (Fig. 5b and Supplementary Figure 26). In addition, the ThT results showed that, in spite of the fast kinetics of fibril formation for both subunit proteins compared with CsgA, CB-origami accelerated fibril formation of the R5 subunit rather than R1 (Fig. 5c). Specifically, $\tau 50$ for R5 was reduced by 1.5 ± 0.3 h in the presence of CB-origami while the value remained almost the same for R1 either in the presence or absence of CB-origami. Moreover, kinetic analysis of the ThT assay via a global fitting demonstrated that the $\lambda$ value for R5 subunit slightly increased from 0.30 ± 0.01 h$^{-1}$ to 0.33 ± 0.02 h$^{-1}$ when R5 was added to CB-origami while R1 subunit remained almost unchanged. These results thus implied that R5 domain was more aggregation-prone than R1 and more strongly contributed to CsgA polymerization, which was consistent with the hypothesis previously proposed by the Chapman group[47].

Functional amyloids with decorated domains/moieties have tremendous potential for applications in nano-science and materials engineering[34,48–50]. To explore whether CB-origami induces the polymerization of functional amyloids, we applied CB-origami as an organic template to site-specifically guide the assembly of functional CsgA proteins (Fig. 5d). We rationally constructed functional CsgA proteins, CsgA-spytag and CsgA-CBD, through the genetic fusion of two functional domains at the C-terminus of CsgA: spytag, a short peptide with 13 amino acids, and chitin-binding domain (CBD), a small protein domain with 45 amino acids. Specifically, spytag covalently reacts with its partner protein spycatcher, while CBD recognizes and binds to chitin[51]. As expected, fibrils grew from the designated position of CB-origami (Fig. 5e). These tethered fibrils had an average height of 3.8 ± 0.5 and 4.6 ± 0.6 nm for CsgA-spytag and CsgA-CBD, respectively (Supplementary Figure 26). Moreover, the ThT

results showed that $\tau 50$ for CsgA-spytag decreased from 50.6 ± 1.3 to 35.3 ± 1.3 h, and for CsgA-CBD from 56.3 ± 0.6 to 40.2 ± 0.9 h (Fig. 5f). In addition, analysis of the kinetics of these reactions revealed that the addition of CB-origami to functional CsgA solutions significantly increased their primary polymerization rate, $\lambda$, implying an accelerated nucleation role of CB-origami in the assembly of functional CsgA proteins. Taken together, these data thus confirmed that CB-origami could serve as nucleators to accelerate functional CsgA polymerization and guide the assembly of functional amyloid proteins in a precisely site-specific fashion.

**Diverse DNA origami/amyloid complex structures**. DNA origami/protein hybrid structures have been of great interest because they have a wide range of potential applications. In this respect, the nucleation-directed assembly of amyloid proteins within DNA origami in a site-specific manner provides a promising approach for constructing DNA origami/protein hybrid structures. We designed diverse CB-origami structures with variable anchoring positions for CsgB and imaged corresponding DNA/protein hybrid structures using AFM under air conditions (Fig. 6a and Supplementary Figure 27). When CsgA monomer proteins were incubated in the presence of CB-origami, fibrils nucleated and grew from the pre-designated position, as expected. In addition, two or three pieces of individual fibrils could be assembled and arranged along the origami by applying CB-origami designed with two or three CsgB nucleators already positioned at pre-defined sites along the edge (Fig. 6a and Supplementary Figure 27). Finally, by incubating the CB-origami structures bearing anchored CsgB nucleators at the three vertexes with a solution of CsgA monomers (2.0 μM), we could assemble complex structures comprising origami and amyloid fibrils. Typically, the structures thus assembled included separated origami connected by individual fibrils or dense complex networks composed of aggregated origami and fibrils (Fig. 6b and Supplementary Figs. 27 and 28).

Notably, all the AFM images collected in the Fig. 6 were carried out in air mode, the AFM imaging in dry condition and possible dehydration during sample preparation, unfortunately, have led to varied morphologies of fibril threads in different AFM images. Nevertheless, these AFM images indeed demonstrated various fibril/DNA origamis structures could be constructed other than the apically labeled origamis by taking advantage of the nucleation role of CsgB protein (Fig. 6a, b and Supplementary Figs. 27 and 28). These interesting results, along with our other findings about the dual roles of CsgB as nucleation and trap sites,

raise important questions that should lead to a better understanding of the specific functional roles that CsgB plays in *E. coli* biofilm production and physiology.

## Discussion

In our study, we applied an approach of CsgA monomer storage in formic acid, which might reduce polymerization kinetics and require the relatively higher concentrations of CsgA for CB-origami-triggered CsgA polymerization. However, as we indeed applied the same protocol/prep/stock methods for CsgA monomers throughout the whole study, therefore, the major conclusions concerning the relative effects (e.g. origami vs. no origami) were still valid in our paper.

In addition, the construction of CB-origami through decoration of CsgB to DNA origami via Ni-NTA metal coordination chemistry was not ideal in our study given the potential replacement reaction between CB-origami and CsgA in solution. However, the ThT results in Fig. 3b, along with nucleation efficiency comparison (Fig. 2b) and real-time AFM imaging (Fig. 4), could indeed lead to solid conclusions concerning relative effects (e.g. origami vs. no origami) in triggering CsgA polymerization. We therefore conclude that the designed DNA origami nucleators, CB-origami, can stimulate fibril growth from the DNA origami substrate.

In summary, the DNA origami technique we applied here provides a simple and convenient template to site-specifically anchor CsgB, making it possible to study and even directly visualize with HS-AFM the nucleation-directed polymerization of CsgA in the presence of well-defined molecular nucleators. DNA origami acts as a set of molecular landmarks to accurately localize the positions of individual nucleators and, therefore, makes it possible to inspect and distinguish independent stochastic nucleation events even from an ensemble measurement. Essentially, our technique rules out possible interference caused by molecular self-polymerization, typically encountered in in vitro nucleator-seeded polymerization experiments. Given the widespread presence of molecular nucleators in biology and the common challenge in probing the nucleation processes and molecular mechanisms of such proteins, our platform provides a useful and generalizable method to study the primary molecular nucleation events of protein nucleators. In addition, this technique can also be applied to assess the molecular-scale nucleation and aggregation of both disease-relevant and functional amyloid proteins in biology. Finally, the directional assembly of amyloid fibrils precisely guided by CB-origami provides a promising approach to creating DNA origami/functional amyloid complex structures. Given the demonstrated applications of both DNA origami and amyloid assemblies in nanotechnology, these hybrid structures, combining the programmable features of DNA origami with the diverse and tunable properties of functional amyloids may open up applications for bionanotechnologies.

## Methods

**Gene construction and sequencing**. The genes for CgA, CBD, and spytag were separately amplified by PCR with introduced compatible overhangs for Gibson assembly. Recombinant genes (CsgA, CsgA-CBD, and CsgA-spytag) were constructed using isothermal Gibson assembly and cloned into pET-22b expression vectors. All gene constructs were sequence verified by Genewiz, and the sequencing results for all genes are presented in Supplementary Figure 2–5.

**Purification and characterization of CsgA and derivatives**. Detailed information about protein expression and purification is described in Supplementary method. Purified proteins were assayed with SDS-PAGE and western blotting. Western blots were probed by primary anti-His mouse monoclonal antibody (TransGen, HT501-02) at a dilution of 1:5000. Secondary goat anti-mouse antibodies lgG conjugated to horseradish peroxidase (HRP) (TransGen, HS201-01) was used at a dilution of 1:5000. The specific experimental protocols are described in detail in Supplementary method.

**Design and preparation of DNA origami and CB-origami**. DNA capturing stands were first constructed by conjugation reaction between Maleimido-C3-NTA and Thio-modified capture DNA strands. To design DNA origami with multiple anchoring sites of CsgB at specific positions, certain staple DNA strands in the DNA origami was first replaced with DNA capturing strands in a site-defined manner. The experimental protocol basically followed the design originally proposed by Rothemund[23], and are described in detail in Supplementary method 1.4.

The obtained NTA-origami (5 nM) decorated with NTA-DNA strand was then incubated with CsgB-His proteins and NiCl$_2$ solution for over 1 h to form CB-origami. The resultant solution was subsequently purified to remove the excessive CsgB and Ni$^{2+}$. A similar procedure was applied to produce CA-origami. Detailed protocols are described in Supplementary method.

**Thioflavin T (ThT) assay**. Purified proteins (5.0 μM) were loaded on 96-well black plates with transparent bottoms. Dependent on the specific experiments, the purified protein solution was added either with or without CB-origami (10 nM). ThT was added with a final concentration of 20 μM. Fluorescence was measured every 3 min after shaking 5 s by a BioTek Synergy H1 Microplate Reader using BioTek GEN5 software set to 438 nm excitation and 495 nm emission with a 475-nm cutoff at 25 °C[52].

**Atomic force microscopy imaging**. Samples were first deposited on mica surfaces. atomic force microscopy (AFM) images were then taken in tapping mode either in fluid mode or air mode on a MFP-3D AFM (Asylum Research) using TR400PSA tips (Olympus). Typical scanning parameters were: scan rate = 1–2 Hz, lines = 512, amplitude set point = 150–300 mV, drive amplitude = 180–300 mV, and integral gain = 18.

High-speed atomic force microscopy (HS-AFM) imaging was carried out with Cypher VRS (Asylum Research) using AC10DS tips (Olympus). Typical scanning parameters were: scan rate = 30 Hz, lines = 256, amplitude set point = 150–300 mV, drive amplitude = 180–300 mV, and integral gain = 18.

**Image processing of CsgA fibrillation monitored by AFM**. A kymograph was constructed based on a recorded video timelapse of in situ CsgA fibrillation. The image stack was resliced along a segmented line selection ($y_0$ to $y_{end}$) that follows the growth trajectory of the fibril as determined from the last image. At each time point (i), the pixel values along the fiber trajectory were linearized into a column. Columns were then stacked together to construct a kymograph that represented pixel evolution in space and time.

**Transmission electron microscopy imaging**. Bright-field transmission electron microscopy (TEM) images were collected on an FEI Tecnai Spirit transmission electron microscope operating at 120 kV in bright field mode after staining the samples with uranyl-acetate or binding with gold nanoparticles.

**Global Analysis of experimental kinetic data**. The global fit for the two parameters, $\lambda$ and $\kappa$ (shown in Fig. 2e), was performed using the analytical rate law equation, which is described in detail in Supplementary method. As the predominant mechanism for CsgA polymerization follows secondary pathways (in which the kinetic rate for secondary pathway is significantly higher than that for Primary pathway), in particular, via a monomer concentration-dependent secondary nucleation process, the value of $n_2$ and $n_c$ in Supplementary Eqn 6 and 7 should be 2[41,53]. All global analytical fits were carried out using a Levenberg–Marquardt algorithm.

**Reporting Summary**. Further information on experimental design is available in the Nature Research Reporting Summary linked to this article.

## Data Availability

The main data supporting the findings of this study are available within the article and its Supplementary Information files. Extra data are available from the corresponding author upon reasonable request. The raw data underlying Figs. 2b, 3c, 4d–e, and 5c, f, Supplementary Figures 6b, 8a, b, 14, and 24, and Supplementary Tables 1 and 2 are provided as a Source Data file.

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

## Acknowledgements

We thank Xinyan Wang of the Analytical Instrumentation Center (AIC) at School of Physical Science and Technology (SPST), ShanghaiTech University, Weiyan Liu, Yilan Jiang of Center of High-resolution Electron Microscopy, SPST, ShanghaiTech University (Grant No. 02161943), and Xiaoxu Tian and Dr. Chao Peng of the Mass Spectrometry System at the National Facility for Protein Science in Shanghai (NFPS), Zhangjiang Lab, China for providing technical support and assistance in data collection and analysis. This work was supported by the Joint Funds of the National Natural Science Foundation of China (Seed Grant No. U1532127) and National Natural Science Foundation of China (Grant No. 31570972), the Dawn Program of Shanghai Education Commission, China (Grant No. 14SG56) for C.Z., China Postdoctoral Science Foundation Grant, China (Grant No. 2016M601682) for X.M.; C.Z. also acknowledges start-up funding support from ShanghaiTech University and 1000 Youth Talents Program, granted by the Chinese Central Government. X.M., J.L., L.W., and C.F. acknowledge the support from NSFC (U1532119, 21675167, 21834007, and 21804088).

## Author contributions

C.Z. and C.F. directed the project. X.M. conceived the technical details and designed the experiments. X.M. carried out ThT, AFM, and TEM experiments. K.L. contributed to the gene cloning and provided CsgA and CsgB monomers. M.L. and X.M. analyzed the ThT data and HS-AFM. X.W. provided gold nanoparticles. X.M. and C.Z. wrote the manuscript with help from all the authors. X.M., K.L., M.L., X.W., T.Z., B.A., M.C., Y.L., J.P., J.L., L.W., T.L., C.F. and C.Z. revised the manuscript.

**Additional information**

**Competing interests:** The authors declare no competing interests

