## [Peer Review File · Nature Communications]

Reviewers' comments:

Reviewer #1 (Remarks to the Author):

This paper deals with the controlled growth of functional E. coli-derived amyloid fibrils from DNA origami nanostructures. To this end, DNA origami triangles were modified to carry CsgB nucleator proteins which induced the site-specific nucleation and subsequent fibrillization of CsgA proteins. Fibrillization was followed in-situ using high-speed AFM. DNA origami-attached CsgB proteins were found to act both as fibrillization nucleators and terminators. Introduction of multiple CsgB proteins at different sites furthermore allowed the assembly of hierarchical DNA origami-amyloid hybrid complexes and networks.

This work is of high relevance as it not only combines two different self-assembly approaches but also demonstrates the fabrication of DNA-protein hybrid nanostructures. Especially the latter topic has recently received growing attention. The experiments are well designed, the presented data appear sound and original, and the conclusions seem well justified. The scholarly presentation is excellent. Nevertheless, some improvements are required prior to publication:

- While the results of the ThT assay appear convincing, an additional control experiment is needed for further verification. ThT is known to interact not only with beta-sheet rich protein aggregates but also with DNA which also affects its fluorescence properties, see e.g. DOIs 10.1016/S1010-6030(03)00320-4, 10.1039/c4cp02838d, and 10.1002/qua.25349. Therefore, I ask the authors to provide ThT fluorescence data in the presence of DNA origami only to establish the level of background emission over the time course of the assay. Furthermore, ThT is known to act as a photosensitizer, especially when bound to DNA (see DOI 10.1016/0047-2670(81)85379-8). Therefore, I'd also like to see AFM images of the DNA origami after the ThT assay to verify them remaining intact for the duration of the assay.

- Fig. 4d,e: It seems that the histograms of elongation distance and stagnation time were fit with Gaussians while the data in the rhs panels are presented as simple mean values. Since the histograms are somewhat asymmetric, however, I ask the authors to rather present and compare the results of the fits, i.e. peak positions and FWHM values. This is particularly important for the stagnation time histogram of the independent mode in Fig. 4e which shows two rather broad peaks, one centered at about 2.3 min and one centered at below 1 min, the latter being much closer to the peaks observed for the departure and arrival modes than the depicted mean values would suggest.

Minor comments:

- The TEM image in Fig. 6b is missing a scale bar.
- Refs. 26 and 30 are identical.

Reviewer #2 (Remarks to the Author):

Mao et al. report on a method aimed at allowing spatial control of curli nucleation, using DNA origami modified with the curli nucleator CsgB.

Curli are bacterial functional amyloids that have gained considerable interest as self-assembling biomaterial in various biotech applications. Although the curli subunit CsgA will spontaneously nucleate to form amyloid fibrils, the process is expedited by the minor curli subunit CsgB. Either way, nucleation is a stochastic process that is difficult to control. The ability to localize nucleation using CsgB DNA origami is interesting both for technological application as well as from a fundamental perspective for the study of amyloid nucleation phenomena.

Although of high interest in principle, the reported experiments and the conclusions drawn have several shortcomings that need to be clarified.

Major points:

- A. The experiments meant to demonstrate association of curli with CB-origami vertexes and specificity of CB-origami nucleation are not unambiguous at present.
- B. The departure and arrival of CsgA fibers from or to CB-origami and associated growth kinetics require additional experimental support.
- C. In several places, conclusions are speculative and/or not supported by the observations

Specific points:

1. Lines 114-122, Fig 2. and Supplementary Fig. 10.

The concentration series in Suppl. Fig. 10 does not show convincing nucleation originating from the DNA origami. Instead, fibers appear to be randomly deposited over the images surfaces. Also the TEM image shown in Fig. 2d does not convincingly show the presence of a CsgA fiber tethered to the CB-origami. This would be needed to demonstrate that fibers are indeed associated with CB-origami.

Also, how do the authors explain the sharp concentration dependence in fiber extension, i.e. fiber growth at 2 μM , whilst at 1 μM it is argued that CsgA nucleates on CB-origami, but does not extend into fibrils? Is this a thermodynamic or kinetic effect? This narrow concentration dependence is unexplained and is in contrast with previous reports that show curli fiber nucleation and growth occur even at low nanomolar concentrations (Sleutel et al. 2017). The low nucleation and elongation efficiency is worrying. The CsgA used originates from monomers depolymerized from fibers using formic acid. Do the authors now that the purified material retains 100% polymerization efficiency? Formic acid and also oxidation can lead to modified CsgA that is strongly compromised in fiber formation (for example Wang et al. DOI: 10.1021/acs.biochem.7b00241). The high concentrations of CsgA needed for fiber formation and the slow growth kinetics compared to previous studies may suggest that only a small fraction of the purified material is still active.

What is the basis to argue that CB-origami mediated nucleation of CsgA is saturated at 3 μM ? No data is shown backing up this claim. This would be a highly unexpectedly narrow dynamic range for nucleation and elongation.

2. Line 125-128. The experiment using gold nanoparticles to demonstrate the molecular nature of the fibers seen in AFM is not clear (Supplementary Figure 13). This should be shown using TEM.

3. Ln. 137- 142 and Supplementary Fig. 14 describe an experiment investigating the specificity of CsgA nucleation on CB-origami vs. CA-origami (i.e. 52 and 14%, respectively). Two observations cast a doubt on the nucleation specificity: (i) apart from particles near the origami vertexes, there are many similar-sized particles in the background. At least for a number of the highlighted DNA origami's, it looks as if the proximity to the vertex may be random and dependent on the density of origami and nucleus-like particles. (ii) in the CB-origami, the density of nucleus-like particles in the background seems to have gone up proportionally (i.e. roughly 3 to 4-fold) with those that appear vertex associated. How is this explained?

In Figure 3B, the ThT experiment should include a plot of 5 μM CsgA in presence of CsgB at an equivalent concentration to that present in the DNA origami.

4. The supplementary videos meant to demonstrate departure or arrival of CsgA fibers from or to CB-origami are not unambiguous. The authors should show larger fields rather than close-ups of isolated CB-origami's. As reported by Sleutel et al. CsgA fibers growing on mica stop elongation when bumping into neighbouring fibers. Similarly growing fibers may stop elongating when bumping into origamis on the mica surface. In arrival mode, would a fiber gain directionality towards a origami vertex?

5. Ln. 207 – 214, Fig. 4. The authors show data suggesting that CsgA fibers in departure or arrival mode have different growth kinetics compared to isolated fibers. How do the authors explain that a fiber would have altered growth kinetics prior to termination in CB-origami?

6. In Ln. 230 – 236 the authors suggest that the higher CsgA fiber growth rates in CB-origami vs. randomly nucleated experiments may stem from a higher effective concentration in the former. There is no evidence for the argument that CsgB generates a smaller number of longer polymers. This should be measured. Irrespective of that, when following growth kinetics of fibers forming at early time-points the effective concentrations of CsgA would be equivalent whether nucleated by CB-origami or spontaneously.

7. In Fig. 6b the AFM and TEM images supposedly show CB-origami networks connected via CsgA fibers, spawn from CB-origami vertexes by nucleation and connecting other fiber by arrival mode. It is not clear that the lines connecting DNA origami are CsgA fibers: (i) in all AFM images, CsgA fiber show more intense, i.e. at different height than the DNA, in 6B the opposite is through. (ii) several origami vertexes appear to have multiple apparent fibers originating or arriving, and (iii), the apparent fibers branch and change in width and intensity along their length, unlike what has been previously seen by TEM or AFM. Also, the TEM shown in Fig. 6b does not convincingly show the presence of fibers. CsgA fibrils are easily imaged by negative stain TEM, the presence of DNA-associated fibers should be unambiguously shown by TEM.

8. Ln 308-312. The authors suggest there is in vivo evidence for curli branching, but provide no citation. Presumably the refer to Bian & Normark 1997. The claims of curli branching made in this paper have not been substantiated in later studies.

9. In Methods, point 5, what concentration of DNA NTA is used?

10. In Methods, point 6, it is said that the equivalent of 1 nM CB-origami is uncubated with micromolar range CsgA over night. The kD of Ni-NTA with 6-His tag is roughly 14 nM. At these concentrations, and with such high concentration of His-tagged CsgA, one would expected that most origami-bound CsgB is replaced by CsgA.

11. In Methods, point 15, it is argued that the predominant mechanism for CsgA polymerizations follows secondary pathways. What is this argument based on?

Minor points.

1. There is no point showing the sequencing results, Supplementary Figs. 3 to 5 can be removed.

2. What amounts of CsgA or CsgB are loaded in the SDS-PAGE and Western blots shown in Supplementary Fig. 8? Why are several bands in the pre-stained molecular mass ladder black and overexposed?

3. Supplementary Fig. 12 shows histograms of fiber heights, with counts plotted on the Y-axis. What does "presented as mean \pm s.e.m." mean? There are no error bars shown what does the mean stand for? The same phrase is present for the histograms shown in Supplementary Fig. 17, but also its use in Fig. 2b needs clarification. Where to the errors originate in the latter? From multiple replicates? How many?

4. Not all authors are mentioned in the author contribution description. Please add the contribution of Tianxin Zhao, Bolin An, Mengkui Cui, Yingfeng Li, Jiahua Pu, Jiang Li, Lihua Wang and Timothy

K. Lu.

Reviewer #3 (Remarks to the Author):

This very interesting paper has taken concepts from several key papers in the field and pieced these together to make not one but several advances. The authors grow curli from DNA origami triangles, provide new insights into the mechanisms of fibril growth and develop new functionalised components as well as cross-linked fibril/DNA materials that will likely be useful for future materials development. The paper is a pleasure to read but there are a few areas where the research requires further clarification.

Results and discussion:

If this text is the first description of the DNA origami triangle, which appears to be the case, it would be useful to describe the dimensions of this building block including width, thickness and height when it is first introduced. When the height of the CsgA nanofibril is also discussed from Figure 2a, it would also be useful to compare here the relative heights to the DNA structure and fibril and make this comparison clearer.

It looks like all the curli protein molecules used to grow fibrils in all experiments were C-terminally his tag modified. It would be good to clearly specify this in the main text. The authors establish that the proteins with the his tag form fibrils (i.e. have an X-ray diffraction pattern) but it would be good to further clarify if the addition/position of the His tag has changed any of the characteristics of these fibrils (width, X-ray reflections, FTIR) relative to the naturally occurring system. I don't think it particularly matters if the fibrils are slightly altered – they are still interesting in their own right but if the authors make conclusions about the mechanisms of native curli fibril growth they should be careful to state that the observations are for the His-tag modified system and not the native system, as sequence modification can have a big impact on growth kinetics. A short discussion on whether the His tag is likely to change the fibrils, whether any differences have been established or shown to be insignificant and whether some or none of these his tags are expected to be incorporated into the fibrils would be useful.

'For comparison, we used Csg-A decorated DNA origami.....' – it would be good to add here that the protein was not expected to nucleate with a reference to prior work establishing this behaviour. There are also no page or line numbers in this version of the text.

Are all the videos labelled? And are all referred to within the text? Some attention to labelling and descriptions would be helpful.

The text indicates that there are a smaller number of longer fibrils when CsgB is present compared to a large number of shorter fibrils for random nucleation. The authors suggest that this could be directly related to the CsgA concentration, which may be larger in the case of random nucleation. This sounds like a hypothesis that could be tested mathematically using a modification of the growth kinetics employed. Are the authors able to model and prove this idea? Even in a simplistic way?

After discussion of the data in Figure 5f, the authors speculate that the kinetics of the reaction and polymerisation rate is faster when nucleated by the CB-origami. Why might this be the case? Can you hypothesise why the energetics have changed in this case? What might the mechanism be?

In the supplementary information the FTIR spectra doesn't look typical for fibrils. Can the authors explain why there is a particularly high plateau at wavenumbers higher than 1700 cm⁻¹? Is this a problem with background subtraction? It may also be helpful to identify the peaks with arrows and

specify the key peaks and what features give rise to these vibrations for completeness.

For supplementary Figure 10, it would be good to have the image rearranged so the concentrations increased from left to right, this would make it easier to read. The legend here could also make a statement about how at higher fibril concentrations the DNA origami can't be resolved by this imaging technique as the DNA is under the fibrils- or at least this is what I am assuming is happening here but a line or two of explanation would be helpful.

Supplementary information Figure 11 – it would be useful to put arrows on the trace to indicate the height of the different features.

Supplementary Figure 18 – what is the vertical fibril in the middle of the middle image? I'm not sure I understand what is going on here and an explanation would be useful. Is this another nucleated fibril? I can't see this on the third image on the right.

Aug 15th, 2018

Re: Directing Curli Polymerization with DNA-Origami Nucleators

SUMMARY:

We would first like to thank the editor and the three reviewers for their helpful and constructive input about our manuscript. Before getting into the full point-by-point response (below), we would like to summarize the key aspects of our revision experiments and our redrafting of the manuscript.

1. We used atomic force microscopy (AFM) and transmission electron microscopy (TEM) to reexamine and analyzed the morphology of single fibril at the vertices of DNA origami, confirming that the CsgA fiber tethered to the CB-origami.
2. We have performed additional control experiment (by monitoring the ThT fluorescence with DNA origami alone) to rule out the possible influence of DNA origami on our ThT assay results.
3. We have tested the stability of DNA origami in the presence of ThT by examining the morphology of DNA origami after ThT incubation with AFM imaging.
4. We also performed another AFM image analysis (large area view) of a nucleation/polymerization time course for a single sample (16 sampling times over the course of half hour), allowing us to illustrate the polymerization trajectories of individual fibrils.
5. We have provided additional movie for *in situ* HS-AFM imaging with larger fields to demonstrate arrival mode indeed exist during the CsgA polymerization in the present of CB-origami.
6. We have presented and compared all the peak positions and FWHM values for the distribution curves of both elongation distance and stagnation time in various modes rather than just presenting the simple mean values.
7. We have constructed a kinetic model (mass/time) based on our ThT data to further demonstrate the concentration-dependent kinetic effect of CsgA polymerization in the presence of CB-origami, which implied that CB-origami-induced polymerization was more easily observed at lower CsgA concentration.
8. To address concerns about the CsgA materials from the reviewer, we have carried out additional SDS-Page analysis to provide more convincing evidence that formic-acid-treated CsgA retains near complete polymerization efficiency.
9. To fully estimate the effect of purification on CsgA polymerization, we denatured CsgA with HdhCl and Formic Acid (FA), and then monitored and analyzed both samples using ThT assays. We found that the use of FA-denatured CsgA, compared to GdhCl-denatured CsgA sample, is more appropriate in our experiments as our project focused on the nucleation process of CsgA polymerization and CsgA sample

is preferred used in monomer format from the start.

10. We have provided additional AFM images to confirm that gold nanoparticles were indeed directly anchored to fibrils tethered to the CB-origami.

Regarding revisions to the text, we have carefully softened some of our claims and speculations as guided by the reviewers. In particular, we have removed some speculative ideas and statements from the revised manuscript, such as CsgB-mediated curli branching and the saturated concentration for the CB-origami mediated nucleation of CsgA. We also have used the word ‘trap’ to replace many previous instances of ‘terminator’ and similar phrases. We have also added content as directed, and have corrected several diction and typographical errors.

We deeply appreciate the work that the editor and reviewers have done on our behalf. As we trust you’ll agree, we feel that both the rigor of our study and the quality of our manuscript have obviously improved as a result of following the thoughtful and helpful reviewer input during our revision process. Kindly see our detailed responses to this input in the point-by-point text that follows.

Our point-by-point response:

Reviewer #1 (Remarks to the Author):

This paper deals with the controlled growth of functional E. coli-derived amyloid fibrils from DNA origami nanostructures. To this end, DNA origami triangles were modified to carry CsgB nucleator proteins which induced the site-specific nucleation and subsequent fibrillization of CsgA proteins. Fibrillization was followed in-situ using high-speed AFM. DNA origami-attached CsgB proteins were found to act both as fibrillization nucleators and terminators. Introduction of multiple CsgB proteins at different sites furthermore allowed the assembly of hierarchical DNA origami-amyloid hybrid complexes and networks.

This work is of high relevance as it not only combines two different self-assembly approaches but also demonstrates the fabrication of DNA-protein hybrid nanostructures. Especially the latter topic has recently received growing attention. The experiments are well designed, the presented data appear sound and original, and the conclusions seem well justified. The scholarly presentation is excellent. Nevertheless, some improvements are required prior to publication:

Responses from the authors:

We thank the reviewer for the positive and insightful comments, which significantly helped us to improve our manuscript.

1. While the results of the ThT assay appear convincing, an additional control experiment is needed for further verification. ThT is known to interact not only with beta-sheet rich protein aggregates but also with DNA which also affects its fluorescence properties, see e.g. DOIs 10.1016/S1010-6030(03)00320-4, 10.1039/c4cp02838d, and 10.1002/qua.25349. Therefore, I ask the authors to provide ThT fluorescence data in the presence of DNA origami only to establish the level of background emission over the time course of the assay.

RESPONSE: Following the reviewer's suggestion, we have carried out an additional control experimental to rule out the possible influence of DNA origami on our ThT assay results. As shown in Response Figure I (below), we monitored the ThT fluorescence in the presence of DNA origami and found that the DNA origami did not produce fluorescence; thus, the normalized background emission over the whole period of the experiments is insignificant compared to the signal for CsgA. We have included this new control experiment in the supporting information (Supplementary Figure. 18).

Response Figure I. ThT assay for CsgA only (5 μ M, red curve) and DNA origami only (10 nM, blue curve).

2. Furthermore, ThT is known to act as a photosensitizer, especially when bound to DNA (see DOI 10.1016/0047-2670(81)85379-8). Therefore, I'd also like to see AFM images of the DNA origami after the ThT assay to verify them remaining intact for the duration of the assay.

RESPONSE: Following the reviewer's comments, we performed AFM imaging of the DNA origami after incubation with ThT (20 μ M) for 3 days. As shown in Response Figure II (below), the morphology of the DNA origami did not change, even after a three days' incubation. Our results thus verify that the DNA origami remained intact for the duration of the ThT assay. We have included this data in the supporting information (Supplementary Fig. 19).

Response Figure II. AFM images of typical triangular DNA origami after incubating with ThT for 3

days. The images indicated that DNA origami remained intact for the duration of the ThT assay. Scale bars: 200 nm

3. Fig. 4d,e: It seems that the histograms of elongation distance and stagnation time were fit with Gaussians while the data in the rhs panels are presented as simple mean values. Since the histograms are somewhat asymmetric, however, I ask the authors to rather present and compare the results of the fits, i.e. peak positions and \pm FWHM values. This is particularly important for the stagnation time histogram of the independent mode in Fig. 4e which shows two rather broad peaks, one centered at about 2.3 min and one centered at below 1 min, the latter being much closer to the peaks observed for the departure and arrival modes than the depicted mean values would suggest.

RESPONSE: Following the reviewer's comments, we have presented and compared all the results for the Gaussians fits, and have now included this data in the main text (Page 8 line 31) and in Supplementary Table I and II. Now, having fully compared the peak position and FWHM values for the distribution curves of both elongation distance and stagnation time in various modes (Response Table I & II, below), it is clear that the stagnation time for the departure and arrival modes are almost identical; there are two typical Gaussian peaks for stagnation time in the independent mode of CsgA polymerization. This observation is consistent with the results described by Sleutel *et al.*¹ (2017; *Nature Chemical Biology*) in which they found that the polymerization of isolated CsgA fibrils exhibits two unique growth speeds. Although there are two peaks, the average stagnation time for the independent-a peak and the independent-b peak are both significantly higher than that for the departure or arrival mode peaks, collectively suggesting that a relatively slower growth speed occurs in the independent growth mode. We have added corresponding discussion in the revised manuscript (Page 9, Line 11).

Table S1. Comparison of elongated distance for different elongation mode.

Mode	Peak position (nm)	FWHM (nm)
Departure	4.02±0.32	6.00
Arrival	4.72±0.64	6.07
Stochastic	6.43±0.35	7.69

Response Table I. Comparison of elongated distance for different elongation mode.

Table S2. Comparison of stagnation time for different elongation mode.

Mode	Peak position (min)	FWHM (min)
Departure	0.142±0.005	0.257
Arrival	0.097±0.076	0.369
Stochastic-a	0.434±0.131	0.692
Stochastic-b	2.13±0.122	0.481

Response Table II. Comparison of Stagnation time for different elongation mode.

Minor comments:

- The TEM image in Fig. 6b is missing a scale bar

RESPONSE: Thank you for pointing out this omission; we have now added a scale bar to this image.

- Refs. 26 and 30 are identical.

RESPONSE: Thank you for pointing out this duplication; we have corrected this by deleting Ref. 30.

Reviewer #2 (Remarks to the Author):

Mao et al. report on a method aimed at allowing spatial control of curli nucleation, using DNA origami modified with the curli nucleator CsgB. Curli are bacterial functional amyloids that have gained considerable interest as self-assembling biomaterial in various biotech applications. Although the curli subunit CsgA will spontaneously nucleate to form amyloid fibrils, the process is expedited by the minor curli subunit CsgB. Either way, nucleation is a stochastic process that is difficult to control. The ability to localize nucleation using CsgB DNA origami is interesting both for technological application as well as from a fundamental perspective for the study of amyloid nucleation phenomena. Although of high interest in principle, the reported experiments and the conclusions drawn have several shortcomings that need to be clarified.

Responses from the authors:

We appreciate the constructive and insightful suggestions from the reviewer, which have helped greatly in improving our manuscript.

Major points:

A. The experiments meant to demonstrate association of curli with CB-origami vertexes and specificity of CB-origami nucleation are not unambiguous at present.

RESPONSE: To address the concerns from the reviewer, we have carried out additional TEM and AFM imaging to provide more convincing evidences that CsgA fibers were indeed directly tethered to the CB-origami. We have included these additional data in the Supplementary Fig S14 and Fig S23 (also see in Response Fig. IV & IX). In these experiments, we observed that the departure or arrival mode for nanofiber attachment at the vertex of DNA-origami is not a random event (that is, we did not observe nanofibril originating from random sites of DNA origami other than vertexes). In contrast, for the samples with non-decorated origami structures, we did observe nanofibrils originating from random places rather than vertexes. Finally, we also observed that the attachment of nanofibers to CsgB-origami is site-dependent (that is, origami with two sites often anchored two nanofibrils, those with three anchored three nanofibrils, etc) (Supplementary Fig. 23).

Collectively, these results confirm the specificity of CB-origami nucleation for CsgA polymerization.

B. The departure and arrival of CsgA fibers from or to CB-origami and associated growth kinetics require additional experimental support.

RESPONSE: While we are not precisely sure which type of experiments the reviewer was suggesting, we chose to address this comment by conducting an AFM image analysis of a nucleation/polymerization time course for a single sample (16 sampling times over the course of 1 hour). This series of images allow us to illustrate the polymerization trajectories of individual fibrils (Response Figure III below). We have marked examples of the departure process in white and blue, and the arrival process in black and light purple. In particular, the white and black markers indicate the original states of departure and arrival mode, respectively. These results indeed showed that fibril polymerization followed two modes in the presence of CB-origami, that is, departure and arrival mode. We have included this data in the supporting information (Supplementary Fig. S21).

Response Figure III. Time-lapse imaging of growing fibers at the vertices of CB- origami. The examples (A, B, C, D, E and F) for departure mode for CsgB-mediated CsgA polymerization are marked in blue and white, while the examples (Z, Y, X, W, V and U) for arrival mode are marked in black and light purple. In particular, the white and black markers indicate the initial states of departure and arrival mode, respectively. The subscript number 0 indicates the original state before in situ CsgA polymerization. Scale bars: 200 nm.

C. In several places, conclusions are speculative and/or not supported by the observations.

RESPONSE: We now appreciate that we had overstepped with our interpretations in several cases in the initially submitted manuscript. We have removed or toned down those instances in the revised manuscript, and these changes are detailed in the individual sections below. We have sought to be highly data-driven in all of our interpretations in the revised work, and we trust you will find our speculative content to be more grounded and reasonable in this version.

Specific points:

1. Lines 114-122, Fig 2. and Supplementary Fig. 10. The concentration series in Suppl. Fig. 10 does not show convincing nucleation originating from the DNA origami. Also, the TEM image shown in Fig. 2d does not convincingly show the presence of a CsgA fiber tethered to the CB-origami. This would be needed to demonstrate that fibers are indeed associated with CB-origami.

RESPONSE: To address the concerns from the reviewer, we have carried out additional TEM and AFM imaging to provide more convincing evidences that CsgA fibers tethered to the CB-origami are indeed associated with CB-origami (Supplementary Fig 14 and 23 (also see Response Fig. IV and IX below)). In these TEM images, although the data don't exhibit high imaging resolution like AFM, we did observe bunches of biomolecule aggregates connected to the equilateral triangular shaped structure; examples of the short fibril decorated-origami have been marked in white, indicating that CsgA fibril indeed connect to the vertices of origami.

Moreover, after washing the samples with water or buffer at least 3 times during preparation for the AFM and TEM imaging, the CsgA aggregates remained joined to the vertices of DNA origami, indicating that the aggregates are indeed firmly associated with CB-origami at its vertices.

Collectively, these results confirm that fibrils originate from the DNA origami as well as the specificity of CB-origami nucleation for CsgA polymerization.

Response Figure IV. AFM and TEM images showing that CsgA fibrils were tethered to the origami landmark.

The concentration series in Supplemental Fig. 10 were presented to reveal that CB-origami-induced nucleation events were more easily observed at relatively low concentration of CsgA, as more random nucleation events could occur at higher concentration of CsgA. The previous data were based on AFM imaging of samples after 24h incubation, in which the samples were found covering the mica surfaces. In the new experimental data, we also used AFM to image formed aggregates of CsgA (5 μ M) after incubating with CsgB for relatively short time (6 hours) (Response Figure V below). In this experiment, we observed a lot of self-assembled oligomers and fibrils were deposited on the mica surface, implying that higher CsgA concentration indeed leads to the formation of more self-assembled oligomer and fibrils, which were further deposited on the mica surface. This result is consistent with our latter kinetic model in Response Figure VI (also see Supplementary Fig S11).

Response Figure V. AFM image of typical triangular DNA origami after incubating with 5 μ M CsgA for 6 hours.

Also, how do the authors explain the sharp concentration dependence in fiber extension, i.e. fiber growth at 2 μ M, whilst at 1 μ M it is argued that CsgA nucleates on CB-origami, but does not extend into fibrils? Is this a thermodynamic or kinetic effect? This narrow concentration dependence is unexplained and is in contrast with previous reports that show curli fiber nucleation and growth occur even at low nanomolar concentrations (Sleutel et al. 2017). The low nucleation and elongation efficiency is worrying. The CsgA used originates from monomers depolymerized from fibers using formic acid. Do the authors now that the purified material retains 100% polymerization efficiency? Formic acid and also oxidation can lead to modified CsgA that is strongly compromised in fiber formation (for example Wang et al. DOI:10.1021/acs.biochem.7b00241). The high concentrations of CsgA needed for fiber formation and the slow growth kinetics compared to previous studies may suggest that only a small fraction of the purified material is still active.

RESPONSE: We appreciate the comments and suggestions from the Reviewer 2.

1. Kinetic effects

We speculate that kinetic effects may account for this sharp concentration dependence in fiber extension. According to the model reported by Knowles et al²⁻⁴, the kinetic rate for primary and secondary polymerization are, respectively, $\lambda = (2K_n * K_+ * m_0^2)^{1/2}$ and $\kappa = (2K_2 * K_+ * m_0^3)^{1/2}$ (m_0 is the concentration of soluble free monomer). In particular, the kinetic rate of κ , in which CsgA aggregates are also produced, grows exponentially as the CsgA concentration increased.

To further demonstrate this concentration-dependent kinetic effect, we constructed a kinetic model (mass/time) based on our ThT data. We modeled CsgA polymerization with CB-origami in various CsgA concentrations, as shown in Response Figure VI below and also see supplementary Fig. 11. In particular, we calculated the produced mass of CsgA aggregates after incubating for 1 day, and found the produced mass is 3.09 μM in the 5.0 μM CsgA system (black curve), which is about 60-fold higher than that (0.05 μM) of the 1.0 μM CsgA system (blue curve). Notably, this kinetic model is also consistent with the AFM results in Supplementary Figure S10, in which CB-origami were incubated for 1 day with various concentration of CsgA and CB-origami-induced polymerization were more easily observed at lower concentration. Specifically, at higher concentration, (for example, 5 μM), the polymerization already passed the initial nucleation stage and more independent fibrils formed and covered the surface of the substrate so that at higher fibril concentrations the DNA origami beneath the covered fibrils can't be resolved by AFM imaging.

Response Figure VI. kinetic model (mass/time) for CsgA polymerization with concentration ranging from 1.0 μM to 5.0 μM in the presence of CB-origami. Left: fibril mass vs polymerization time curve; right: fibril fraction vs polymerization time curve. Note: the kinetic model was constructed by applying different concentration of CsgA into the kinetic equation proposed by Dr. Knowles^{2,3} (also see Supplementary method 1.15), with the kinetic parameters acquired from data based on ThT assay.

More importantly, the period for the lag phase, in which less than 10% of the free monomers have self-assembled into aggregates⁴, is also highly associated with the CsgA concentration, and we have marked the lag phase for various CsgA concentrations in grey. After 1 day of incubation, the CsgA polymerization for the 1.0 μM CsgA concentration experiment is still in an early stage of lag phase, indicating that most CsgA aggregates are still in an oligomer form. While in the 2.0 μM CsgA concentration experiment, its polymerization is primarily at the early growth phase after 1 day of incubation, indicating that some aggregates have started to elongate to form fibrils.

Collectively, these results support the kinetic effect of concentration on CsgA polymerization.

2. Material purification.

RESPONSE: We prepared freshly FA-denatured CsgA samples in most of our experiments and usually stored them for a short time (less than 1day), which according to Wang et al. 2017 would cause little change or modify in the structure⁵. Moreover, our later results on fibril extension also demonstrate that our FA-denatured CsgA is capable of self-assembling into fibrils. To further address concerns about the CsgA materials from the reviewer, we have carried out additional SDS-Page analysis to provide more convincing evidence that formic-acid-treated CsgA retains near complete polymerization efficiency (Response Figure VII below). In this experiment, no band for CsgA monomers is observed after incubating 72 hours, clearly demonstrating that the great majority of CsgA are re-denatured and are capable of self-assembling into oligomers and fibrils.

Response Figure VII. Coomassie-stained SDS-PAGE for CsgA.

The paper by *Sleutel et al.* applied *in situ* AFM imaging in their studies and found that nucleation started even at low nanomolar concentrations of CsgA¹. In our experiment for Figure 2b and Supplementary Fig 10 &14, our samples were dropped on the mica surface and incubated for more than 1 day before examination with AFM imaging. We speculate that contact between the cantilever and sample and laser warming during the *in situ* AFM imaging may have increased the local protein concentration and thus accelerated the polymerization rate. This situation was also observed in our *in situ* AFM experiments but not with regular AFM imaging. For example, Small CsgA aggregates started elongating even within the first 30 mins, as shown in the Response Fig III. Therefore, the above results indicated that formic acid didn't cause permanent damage on CsgA monomers since the majority of CsgA eventually polymerize after 3 days.

However, we should also point out here that the two denaturing reagents, HdHCl and Formic acid, indeed caused different effects on the re-folding process of CsgA monomers. To fully estimate the effect of purification on CsgA polymerization, we denatured CsgA with HdHCl and Formic Acid (FA), then we monitored and analyzed both samples using ThT assays, as shown Response figure VIII below. In this ThT assay, we found that the t50 of GdHCl is about 14.1 hour, which is similar to the result of that paper presented by *Sleutel et al.*¹. While the t50 for the FA-denatured CsgA is about 34 hour, which is significantly longer than that for GdHCl. To further interrogate these findings, we also analyzed the ThT

data and calculated the kinetic constants for nucleation (K_n) of FA and GdHCl denatured CsgA, respectively. The K_n value for FA (0.841) is significantly higher than that for GdHCl (0.00539). The low K_n in the case of GdHCl denatured CsgA suggested this sample might not be a good candidate for assessing the nucleation event, as GdHCl-denatured CsgA did not actually denature fully into CsgA monomers and most remained in the oligomer form or remained in the active monomer form. Since our project focused on the nucleation process of CsgA polymerization and the CsgB nucleation mechanism, we prefer that CsgA sample would be used in monomer format. We therefore decide to choose FA-denatured CsgA instead of GdHCl-denatured CsgA sample in our experiments.

Response Figure VIII. ThT assay revealing the kinetics of amyloid formation for Formic and GdHCl denatured CsgA

What is the basis to argue that CB-origami mediated nucleation of CsgA is saturated at 3 μM ? No data is shown backing up this claim. This would be a highly unexpectedly narrow dynamic range for nucleation and elongation.

RESPONSE: We are sorry for the considerable confusion that our initial statements about this topic in the initially submitted manuscript caused. We were actually trying to consider this issue theoretically, but we now understand that it appeared as if we were claiming this as a result. It was not intended as such, and we have removed these ideas from the revised manuscript.

2. Line 125-128. The experiment using gold nanoparticles to demonstrate the molecular nature of the fibers seen in AFM is not clear (Supplementary Figure 13). This should be shown using TEM.

RESPONSE: Following the reviewer’s suggestion, we have carried out additional AFM imaging to provide more convincing evidence that gold nanoparticles were indeed directly anchored to fibrils tethered to the CB-origami. We have included the additional data in the supporting information (Supplementary Fig. 14).

Response Figure IX. AFM images showing that gold nanoparticles specifically bound to CsgA fibrils tethered to the origami landmark. Scale bars: 100 nm.

In addition, we have also attempted to perform TEM imaging to reveal the molecular nature of the fibers (Response Figure X below).

Response Figure X. TEM images of CB-origami incubated with CsgA

However, unlike AFM imaging, the imaging contrast of biomolecules in TEM is quite low while that of AuNPs is much higher, as such, only clear feature of AuNPs was found in the TEM image. Therefore, we only presented AFM images in our supplemental information.

3. Ln. 137- 142 and Supplementary Fig. 14 describe an experiment investigating the specificity of CsgA nucleation on CB-origami vs. CA-origami (i.e. 52 and 14%, respectively). Two observations cast a doubt on the nucleation specificity: (i) apart from particles near the origami vertexes, there are many similar-sized particles in the background. At least for a number of the highlighted DNA origami's, it looks as if the proximity to the vertex may be random and dependent on the density of origami and nucleus-like particles. (ii) in the CB-origami, the density of nucleus-like particles in the background seems to have gone up proportionally (i.e. roughly 3 to 4-fold) with those that appear vertex associated. How is this explained?

RESPONSE: We suspected that these “similar-sized particles” are salt contaminants or impurities from solution precipitated on the surface, so, seeking to reduce or avoid the influence of such background signals, we selected a completely clean region of the mica surface (selection based on AFM imaging, see the clean (Supplementary Fig. 16) vs. impure mica surface images (Response Figure XI)) and used these regions to calculate the nucleation efficiency. This process allowed us to exclude the influence of such impurities on nucleation specificity: we found that no matter whether the mica surfaces contain nucleus-like particles or are free of nucleus-like particles, the nucleation efficiency was repeatable for CsgA and CB-origami or CA-origami.

For example, in a typical AFM image collected for sample containing 50 μ L 1.0 μ M CsgA incubated with CA-origami (1 nM) for 24 hours, the mica surface was covered with a significant amount of these “similar-sized particles”, but we found that the nucleation efficiency of CA-origami was 18.5 ± 5.2 %, which is close to the efficiency (14.3 ± 4.6 %) calculated from data collected on a much cleaner mica surface. We hope that this, viewed alongside our other results demonstrating the specificity of nucleation (e.g., Fig. 4a-c, Fig. 6a, Supplementary Fig. 16 and 23), will be deemed satisfactorily convincing for the specificity of CsgA nucleation on CB-origami.

Response Figure XI. Typical AFM images for CB-origami incubated with 1.0 μ M CsgA within impure mica surface. Scale bar: 500 nm.

In Figure 3B, the ThT experiment should include a plot of 5 μ M CsgA in presence of CsgB at an equivalent concentration to that present in the DNA origami.

RESPONSE: Following the reviewer's suggestion, we have performed additional ThT experiments using 5 μ M CsgA solution in the presence of CsgB (10 nM) at an equivalent concentration to that present in the DNA origami (Response Figure XII below). When we compared the data from this new experiment with the other data for CsgA incubated with CB-origami, we found that CsgB, similar to CB-origami, also accelerated the kinetic rate of CsgA polymerization and both cases have a very similar kinetics of polymerization. We have incorporated the curve corresponding to CsgA polymerization in the presence of CsgB into Figure 3b.

Response Figure XII. ThT assay for 5 μ M CsgA incubated with 10 nM CsgB.

4. The supplementary videos meant to demonstrate departure or arrival of CsgA fibers from or to CB-origami are not unambiguous. The authors should show larger fields rather than close-ups of isolated CB-origami's. As reported by Sleutel et al. CsgA fibers growing on mica stop elongation when bumping into neighbouring fibers. Similarly growing fibers may stop elongating when bumping into origamis on the mica surface. In arrival mode, would a fiber gain directionality towards a origami vertex?

RESPONSE: Following the reviewer's suggestion, we have now included the larger fields of AFM in the Supplementary video 5. The original close-ups of the AFM images were just representatives of typical observations based on our multiple experiments. These expanded video scales show, for elongating single fibrils, that the departure and arrival modes are not random events, highlighting the role CsgB in CsgA polymerization. Although we cannot rule out the possibility that growing fibers may stop elongating when they simply bump into the DNA portion of origami structures or into the mica surface, monitoring of growing fibers, along with captured AFM images showed in Response Figure III, showed that they appear to preferentially terminate at vertices rather than other sites of the origami. Mechanistically, as we now detail in the revised text, we speculate that the CsgB on CB-origami can function as traps to capture nanofibrils in their vicinity (see points Response Figure III).

5. Ln. 207 – 214, Fig. 4. The authors show data suggesting that CsgA fibers in departure or arrival mode have different growth kinetics compared to isolated fibers. How do the authors explain that a fiber would have altered growth kinetics prior to termination in CB-origami?

RESPONSE: This is an excellent question, and we speculate that local concentration effects may account for differential kinetics in a single reaction system: at least two factors could alter the growth kinetics of CsgA prior to termination in CB-origami. First, and highly similar to what is known to occur during secondary polymerization of natural amyloid nanofibrils (in which the surfaces of existing fibrils catalyze the nucleation of new aggregates from the monomeric state, with a rate dependent on both the concentration of monomers and that of existing fibrils.), the presence of CsgB may promote the folding of the unfolded “native” CsgA monomers into folded CsgA monomers. A second factor would be a consequence of this promoting effect: in the immediate vicinity of any CsgB molecules there would be an increase in the local concentration of CsgA monomers with β -strands. As such strand structures are prerequisite for polymerization, this could help explain the proposed difference in kinetics.

6. In Ln. 230 – 236 the authors suggest that the higher CsgA fiber growth rates in CB-origami vs. randomly nucleated experiments may stem from a higher effective concentration in the former. There is no evidence for the argument that CsgB generates a smaller number of longer polymers. This should be measured. Irrespective of that, when following growth kinetics of fibers forming at early time-points the effective concentrations of CsgA would be equivalent whether nucleated by CB-origami or spontaneously.

RESPONSE: We appreciate the reviewer’s insightful points. Indeed, we think at earlier stage, the nucleation roles of CsgB may provide dominant roles in affecting CsgA polymerization probably because of stronger molecular interactions via specific molecular recognition, and indeed effective concentration of CsgA would be equivalent whether nucleated by CB-origami or spontaneously at that time. This argument was supported by the evidence that CsgB-origami tends to induce the formation of more oligomers at the vertex compared to CsgA-origami after incubation for 24 hours.

The appearance of oligomers at the vertex of CB-origami would certainly provide less energy barrier for further polymerization of remaining CsgA compared to that of the independent polymerization of CsgA in the absence of any nucleators. In such situation, it is reasonable to assume that polymerization more preferably occurs at the formed CB-origami in early stage, thus leading to a smaller number of longer polymers. To further address the reviewer’s concern, we constructed a kinetic model (the number of produced aggregates/time) based on our ThT data. We modeled CsgA polymerization both with and without CB-origami, as shown in Response Figure XIII below. Addition of CsgB indeed reduced the final number of CsgA aggregates, indicating that more CsgA monomers join the extant fibril instead of self-nucleated into small aggregates. However, as we had originally pushed this the claim too far and as we do not have sufficient data to appropriately support this claim, we have withdrawn these statements from the revised text.

Response Figure XIII. kinetic model (aggregates numbers/time) for CsgA polymerization with 5.0 μM concentration with and without CB-origami.

7. In Fig. 6b the AFM and TEM images supposedly show CB-origami networks connected via CsgA fibers, spawn from CB-origami vertexes by nucleation and connecting other fiber by arrival mode. It is not clear that the lines connecting DNA origami are CsgA fibers: (i) in all AFM images, CsgA fiber show more intense, i.e. at different height than the DNA, in 6B the opposite is through. (ii) several origami vertexes appear to have multiple apparent fibers originating or arriving, and (iii), the apparent fibers branch and change in width and intensity along their length, unlike what has been previously seen by TEM or AFM. Also, the TEM shown in Fig. 6b does not convincingly show the presence of fibers. CsgA fibrils are easily imaged by negative stain TEM, the presence of DNA-associated fibers should be unambiguously shown by TEM.

RESPONSE: Following the reviewer's suggestion, we have performed additional AFM and TEM imaging, which showed CB-origami networks connected via CsgA fibers (Response Figure XIV). We have included this data in the supporting information (Supplementary Fig. 23).

Response Figure XIV. Typical DNA origami-fibril complex structures formed by polymerization of CsgA monomers (2.0 μM) in the presence of CB-origami, with CsgB tethered at the three vertexes. Scale: 100nm

We should point out that CB-origami applied here (Fig. 6b) bear anchored CsgB nucleators at the three vertexes, in contrast with the single CsgB nucleator anchored at one of the three vertexes. Polymerization of CsgA in the presence of CB-origami with triple nucleators led to the formation of DNA origami-fibril complex networks structures that appear to have multiple apparent fibers originating or arriving. These morphological features are indeed different from those of the single nucleator-induced polymerization as shown in Figure 2.

The difference in the intensity and width for CsgA fibril may arise from the different sample prepared and applied in AFM imaging. In figure 2, AFM imaging was purposely carried out under aqueous phase using tapping mode in order to capture the nucleation phase at earlier stage. The height image of AFM imaging in aqueous phase reflects the actual height of DNA origami and CsgA fibrils. However, in figure 6, our major aim was to confirm if the final assembled hybrid structures are nucleation-dependent or not. We therefore have applied AFM imaging in air phase based on tapping

mode, which is a much easier and quicker AFM imaging approach compared to the liquid phase contact mode imaging. As such, the height measured in air may not reflect the actual height of the DNA origami and CsgA fibrils possibly due to sample shrink because of dehydration. In addition, the different tips, such as its sharpness and spring constants, may, to some extent, lead to deflection in the height and intensity of the measured samples.

To show the potential influence of different imaging mode, we compared the height of fibril in air and liquid phase, respectively. (Response Figure XV). This data clearly indicated that the height of fibril in air mode (0.8 nm) is significantly lower compared to the height of fibrils imaged under liquid mode (3 nm). Collectively,

Response Figure XV. Comparison of the height of CsgA fibril in air mode and liquid mode, respectively.

Finally, following the reviewer's suggestion, we have also attempted to apply TEM to collect more convincing data to show the presence of fibers (Fig. 6b and Response Figure XIV). However, we should point out that the complex network structures, shown in AFM images, directly formed on mica surfaces. As such, it was quite easy to find images showing the fibril growing out or towards the DNA origami. In contrast, in TEM sample preparation, the DNA/fibrils structures were first formed in solution and then dropped onto TEM grids, the long fibrils connecting the origami might easily fold during sample preparation and the DNA origami may not well separate from each other during in situ polymerization, making it challenging to obtain high-quality complex network structures that clearly show the presence of fibers.

8. Ln 308-312. The authors suggest there is in vivo evidence for curli branching, but provide no citation. Presumably the refer to Bian & Normark 1997. The claims of curli branching made in this paper have not been substantiated in later studies.

RESPONSE: We appreciate the reviewer's insightful points. We agreed with the reviewer here that the claims of curli branching were based on only one reference and no later studies supported such claims. For these reasons, we had originally pushed this the claim too far and as we do not have sufficient data to appropriately support this claim, we have removed these statements from the revised text.

9. In Methods, point 5, what concentration of DNA NTA is used?

RESPONSE: As mentioned in the Supplementary method 1.14, to ensure the origami is fully modified with NTA-DNA, we incubated 5 nM M13mp18 DNA with 100 nM DNA-NTA, while the concentration of staples DNA is 50 nM. The fresh formed NTA-origami were then applied for further conjugation. Finally, we directly incubated 5 nM freshly made DNA origami containing 100 nM NTA-DNA in solution with 1 μ M CsgB and 5 μ M Ni²⁺. We have added a statement to the Supplementary method 1.5.

10. In Methods, point 6, it is said that the equivalent of 1 nM CB-origami is uncubated with micromolar range CsgA over night. The kD of Ni-NTA with 6-His tag is roughly 14 nM. At these concentrations, and with such high concentration of His-tagged CsgA, one would expected that most origami-bound CsgB is replaced by CsgA.

RESPONSE: The molecular bonds between His-Tag and Ni-NTA is very strong, even 1 mM imidazole can't dissociate them (*Hochuli et al Nat. Bio. 1988*), as evidenced by the requirement of applying high concentration of imidazole to elute His-tagged protein from Ni-NTA resins in typical protein purification protocols. Therefore, we don't think origami-bound CsgB could be replaced by CsgA in the presence of 14 nM CsgA. Meanwhile, the CB-origami was deposited on the mica surface, making it even harder for CsgA to replace them.

11. In Methods, point 15, it is argued that the predominant mechanism for CsgA polymerizations follows secondary pathways. What is this argument based on?

RESPONSE: According to the model reported by Dr. Knowles et al^{2,3}, the kinetic rate for primary and secondary polymerization are, respectively $\lambda = (2K_n * K_+ * m_0^2)^{1/2}$ and $\kappa = (2K_2 * K_+ * m_0^3)^{1/2}$ (m_0 is the concentration of soluble free monomer). By analyzing the ThT data, we calculated the λ and κ for CsgA polymerization (5.0 μ M), respectively. In this result, we found that the κ for CsgA polymerization is about 0.16, which is significantly higher (12-fold) than that (0.013) of λ , indicating that the predominant mechanism for CsgA polymerizations follows secondary pathways. We have added a statement to the Supplementary method 1.15.

Minor points.

1. There is no point showing the sequencing results, Supplementary Figs. 3 to 5 can be removed.

RESPONSE: We highly appreciate the comments and suggestions from the Reviewer 2. These sequencing results are strong evidence to demonstrate that the purified protein is CsgA (with no mutation in the sequence). In the meantime, they were just listed in the supplementary figs. rather than in the main text. The supplemental information can serve as a future reference for those people who are not familiar with but interested in using CsgA in their research.

2. What amounts of CsgA or CsgB are loaded in the SDS-PAGE and Western blots shown in Supplementary Fig. 8? Why are several bands in the pre-stained molecular mass ladder black and overexposed?

RESPONSE: Following the reviewer's suggestion, we have performed additional SDS-PAGE and Western blot imaging, which showed clear single band of purified CsgA and CsgB proteins. We have included the data in the revised manuscript (Supplementary Fig. 8).

Response Figure XVI. Purification and biological assays of purified CsgA and CsgB proteins. (a) & (b)Coomassie-stained SDS-PAGE and western blots with anti-His antibodies confirm the expressed proteins: CsgA (a) and CsgB (b); (c) ThT assay revealing the kinetics of amyloid formation for CsgA. kinetic model (aggregates numbers/time) for CsgA polymerization with 5.0 μ M concentration.

3. Supplementary Fig. 12 shows histograms of fiber heights, with counts plotted on the Y-axis. What does “presented as mean \pm s.e.m.” mean? There are no error bars shown what does the mean stand for? The same phrase is present for the histograms shown in Supplementary Fig. 17, but also its use in Fig. 2b needs clarification. Where to the errors originate in the latter? From multiple replicates? How many?

RESPONSE: “mean \pm s.e.m.” refers to “mean value \pm standard error of mean (s. e. m.)”. Following the reviewer's suggestion, we have specified the above information in the figure caption or in the supplementary figure caption now.

4. Not all authors are mentioned in the author contribution description. Please add the contribution of Tianxin Zhao, Bolin An, Mengkui Cui, Yingfeng Li, Jiahua Pu, Jiang Li, Lihua Wang and Timothy K. Lu.

RESPONSE: We have noted their specific contributions in the author contribution description. We have provided the information here as a reference.

C.Z. and C.F. directed the project. X.M. conceived the technical details and designed the experiments. X.M. carried out ThT, AFM and TEM experiments. M.L. and X.M. analyzed the ThT data and HS-AFM. K.L., T.Z. and M.C. contributed to the gene cloning and plasmid construction. K.L. and B.A. contributed to the expression and purification of CsgA and CsgB monomers. K.L., J.P. and Y.L. performed X-ray fiber diffraction and FTIR for CsgA fibrils. X.W. synthesized Ni-NTA decorated gold nanoparticles. J.L., L.W. and T. K. L provided helpful discussions on the AFM and ThT data. X.M. and C.Z. wrote the manuscript with help from all the authors. All the authors revised the manuscript.

Reviewer #3 (Remarks to the Author):

This very interesting paper has taken concepts from several key papers in the field and pieced these together to make not one but several advances. The authors grow curli from DNA origami triangles, provide new insights into the mechanisms of fibril growth and develop new functionalised components as well as cross-linked fibril/DNA materials that will likely be useful for future materials development. The paper is a pleasure to read but there are a few areas where the research requires further clarification.

Responses from the authors:

Thank you for the positive and encouraging comments, for the constructive suggestions, and for supporting the publication of our work. In our revised manuscript, we have addressed all of the concerns and issues raised Reviewer #3. We firmly believe that the quality of the manuscript has been improved significantly as a result of undertaking these revisions.

1. If this text is the first description of the DNA origami triangle, which appears to be the case, it would be useful to describe the dimensions of this building block including width, thickness and height when it is first introduced. When the height of the CsgA nanofibril is also discussed from Figure 2a, it would also be useful to compare here the relative heights to the DNA structure and fibril and make this comparison clearer.

RESPONSE: We have now added information in the revised manuscript about the dimensions of the triangular DNA origami, including width and height (DNA origami: p4 line 34). Moreover, we have also compared the relative heights of the CsgA nanofibrils with the new DNA origami building block, and we highlight that the height of the nanofibrils is significantly larger than that of the DNA origami. A practical purport of this height difference is that *in situ* AFM imaging is clearly an appropriate technology to use for distinguishing these materials.

2. It looks like all the curli protein molecules used to grow fibrils in all experiments were C-terminally his tag modified. It would be good to clearly specify this in the main text. The authors establish that the proteins with the his tag form fibrils (i.e. have an X-ray diffraction pattern) but it would be good to further clarify if the addition/position of the His tag has changed any of the characteristics of these fibrils (width, X-ray reflections, FTIR) relative to the naturally occurring system. I don't think it particularly matters if the fibrils are slightly altered – they are still interesting in their own right but if the authors make conclusions about the mechanisms of native curli fibril growth they should be careful to state that the observations are for the His-tag modified system and not the native system, as sequence modification can have a big impact on growth kinetics. A short discussion on whether the His tag is likely to change the fibrils, whether any differences have been established or shown to be insignificant and whether some or none of these his tags are expected to be incorporated into the fibrils would be useful.

RESPONSE: Following Reviewer #3's suggestion, we have made the following changes in our revised manuscript:

1. We have clearly specified—in both the main text (P 5, line 4) and the supplementary information (Method 1.2)—that the CsgA molecules used to grow fibrils throughout our experiments were C-terminally poly-histidine tagged.

2. The incorporation of poly-histidine tags onto CsgA molecules to facilitate protein purification is a well-established technique in the Curli amyloid field, in which it is commonly accepted the incorporation of such poly-histidine tags does not obviously disrupt the self-assembly, the CsgA secondary protein structures, or the morphology of the assembled amyloid nanofibrils; we are not aware of rigorous physical studies that have addressed this exact biochemical question specifically, but there are many references reporting a lack of obvious differences between tagged and untagged CsgA structures⁶⁻⁸ (*Chapman et al, Science, 2002*, and *Zhong et al, Nat. Nano, 2014*). Furthering this, we have conducted ongoing studies in which we have not noted obvious differences between tagged and untagged amyloid structures (including CsgA). In the revised manuscript, we have added the following text: “Consistent with several previous studies, our His-Tag fused CsgA proteins had apparently normal self-assembling capacity to form amyloid nanofibrils with typical cross- β structures and that resembled the fibril structures present in wild type biofilms (supplementary Fig. 9).”

3. 'For comparison, we used CsgA- decorated DNA origami.....' – it would be good to add here that the protein was not expected to nucleate with a reference to prior work establishing this behaviour. There are also no page or line numbers in this version of the text.

RESPONSE: Following Reviewer #3's suggestions, we have added a statement to the revised main text to address this concern: "Previous studies have shown that, relative to the influence of CsgB, the presence of free CsgA does not exhibit a significant influence on overall CsgA polymerization, so we here used CsgA-decorated DNA origami (CA-origami) as a control under the same condition (1.0 μ M CsgA solution)."

4. Are all the videos labelled? And are all referred to within the text? Some attention to labelling and descriptions would be helpful.

RESPONSE: Thank you for bringing these inconsistencies to our attention. We have re-labeled these videos and have made sure throughout the text that each is properly referenced. Moreover, we have added brief descriptions about the content of each video in the supporting information (P39, line 1).

5. The text indicates that there are a smaller number of longer fibrils when CsgB is present compared to a large number of shorter fibrils for random nucleation. The authors suggest that this could be directly related to the CsgA concentration, which may be larger in the case of random nucleation. This sounds like a hypothesis that could be tested mathematically using a modification of the growth kinetics employed. Are the authors able to model and prove this idea? Even in a simplistic way?

RESPONSE: Following the reviewer's suggestion, we constructed a kinetic model (mass/time) based on our ThT data (See Response Figure XVII). We modeled CsgA polymerization both with and without CB-origami. The output from the model clearly suggested that CB-origami indeed increased the number of nucleation events, implying the presence of an increased number of sites for nucleation in the reaction system. In such situation, it is reasonable to assume that polymerization more preferably occurs at the formed CB-origami in early stage, thus leading to a smaller number of longer polymers. Addition of CsgB indeed reduced the final number of CsgA aggregates, indicating that more CsgA monomers join the extant fibril instead of self-nucleated into small aggregates.

However, as we had originally pushed this the claim too far and as we do not have sufficient data to appropriately support this claim, we have withdrawn these statements from the revised text.

Response Figure XVII. A kinetic model (aggregates numbers/time) for CsgA polymerization with 5.0 μM concentration with and without CB-origami.

After discussion of the data in Figure 5f, the authors speculate that the kinetics of the reaction and polymerisation rate is faster when nucleated by the CB-origami. Why might this be the case? Can you hypothesise why the energetics have changed in this case? What might the mechanism be?

RESPONSE: Given the content of this comment, we assume that the reviewer was referring to the panel 4f in the original manuscript. In the revised manuscript, we have changed the description of our design concept (reference in figure 1 of the revised text) to reflect our speculation about the energetics driving our observations. We now write in the main text as follows:

“The molecular nucleators were designed by anchoring CsgB proteins onto triangular DNA origami in a position-defined manner. We speculated that the DNA-origami-anchored CsgB would promote the local nucleation rate of CsgA by lowering the energy barrier for β -strand formation of the free CsgA monomers in the vicinity of CsgB, thereby increasing the overall rate of nanofibril polymerization. The resulting directional curli polymerization enabled by CsgB-decorated DNA origami (CB-origami) should closely recapitulate molecular interactions between CsgA and CsgB in *E. coli* biofilms (Fig. 1a).”

Fundamentally we propose that certain feature of the CsgB molecular structure, which promotes the folding and formation of β -strand structures in CsgA (and its presence in the reaction system), is responsible for the increased polymerization rate, typically observed for CsgA polymerization in the presence of our molecular nucleator. We also revisit this mechanistic idea of CsgB increasing the local concentration of CsgA monomers with formed β -strands later in the manuscript (P9, line 11).

In the supplementary information the FTIR spectra doesn't look typical for fibrils. Can the authors explain why there is a particularly high plateau at wavenumbers higher than 1700 cm^{-1} ? Is this a problem with background subtraction? It may also be helpful to identify the peaks with arrows and specify the key peaks and what features give rise to these vibrations for completeness.

RESPONSE: The plateau in the initial figure might result from impurities in our solutions. Upon redoing these experiments in the course of our revision, we did not observe this plateau signal. We have added the new FTIR image of the key peak for the β -sheet in the Supplementary Figure 9 and have labeled it as suggested.

Response Figure XVIII. Morphological and structural characterization of CsgA amyloid fibrils. (a) Representative AFM image of CsgA fibrils; (b) TEM image of CsgA fibrils; (c) FTIR spectrum of CsgA fibrils; (d) X-ray fiber diffraction pattern of CsgA fibrils, a cross- β core structure with an axial reflection at 4.8 \AA and an equatorial reflection at 9.0 \AA arising from the inter-strand and β -sheet spacing. Note: Samples used for characterization were CsgA fibril samples formed after incubation of His-Tag fused CsgA proteins solution for over 48 hours.

For supplementary Figure 10, it would be good to have the image rearranged so the concentrations increased from left to right, this would make it easier to read. The legend here could also make a statement about how at higher fibril concentrations the DNA origami can't be resolved by this imaging technique as the DNA is under the fibrils- or at least this is what I am assuming is happening here but a line or two of explanation would be helpful.

RESPONSE: Following Reviewer #3's suggestion, we have rearranged the panels so that they now represent increasing concentrations from the left to right. (See also Supplemental Fig. 10) We have also accordingly made a statement in the main text and in the figure caption about why, at higher fibril concentrations, the DNA origami cannot be resolved: "and found that the formed CsgA aggregates significantly extended with increased CsgA concentration due to the concentration dominant kinetic effect during CsgA polymerization. To further estimate the nucleation efficiency of CsgB, we further imaged formed CsgA aggregates together with CB-origami in lower concentration of CsgA, such 1.0 μM and 2.0 μM , which apparently is in nucleation phase."

Response Figure XIX. AFM images showing the structures obtained by nucleation-directed polymerization of CsgA solutions of different concentrations in the presence of CB-origami (0.5 nM). The concentrations of CsgA applied (from left to right) were 5.0, 3.0, 2.0 and 1.0 μM respectively. Scale bars: 1 μM . Note: a higher concentration, (for example, 5 μM), the polymerization already passed the initial nucleation stage and more independent fibrils formed and covered the surface of the substrate so that at higher fibril concentrations the DNA origami beneath the covered fibrils can't be resolved by AFM imaging.

Supplementary information Figure 11 – it would be useful to put arrows on the trace to indicate the height of the different features.

RESPONSE: Thanks for helping us to clarify the content of these figures for the readers; we have added arrows on the trace to indicate the height of the different figures. (See also Supplemental Fig. 12).

Response Figure XX. AFM height images and corresponding height traces showing the different heights between the oligomer structures (formed at the vertexes) and DNA origami. The structures were obtained by nucleation-directed polymerization of CsgA solutions (1.0 μ M) in the presence of CB-origami (0.5 nM). Scale bars: 200 nm.

Supplementary Figure 18 – what is the vertical fibril in the middle of the middle image? I’m not sure I understand what is going on here and an explanation would be useful. Is this another nucleated fibril? I can’t see this on the third image on the right.

RESPONSE: To clarify, the second and third panels are not the same samples. For the middle panel, there are two fibrils: the ‘horizontal’ that started from the CsgB nucleator at the vertex and terminates at the CsgB on a second vertex; we strongly suspect that the ‘vertical’ fibril represents an independent aggregation and polymerization ceased when it was trapped by the CsgB on the CB-origami. We speculate that an oligomer deposited on the surface and elongated till it met the fibril between DNA origami.

We should point out that the appearance of the vertical fibril in the middle of the middle AFM image might be an accidental event, as we didn’t find identical structures in other captured AFM images and TEM images (See also Supplemental Fig. 23). Therefore, we decide to provide more AFM images, which showed typical morphology of single fibrils.

Response Figure XXI. Typical DNA origami-fibril complex structures formed by polymerization of CsgA monomers ($2.0 \mu\text{M}$) in the presence of CB-origami, with CsgB tethered at the three vertexes. Scale: 100nm.

References:

1. Sleutel, M. et al. Nucleation and growth of a bacterial functional amyloid at single-fiber resolution. *Nat. Chem. Biol.* (2017).
2. Knowles, T.P.J. et al. An Analytical Solution to the Kinetics of Breakable Filament Assembly. *Science* **326**, 1533-1537 (2009).
3. Cohen, S.I.A. et al. Proliferation of amyloid-beta 42 aggregates occurs through a secondary nucleation mechanism. *Proc. Natl Acad. Sci. USA* **110**, 9758-9763 (2013).
4. Arosio, P., Knowles, T.P.J. & Linse, S. On the lag phase in amyloid fibril formation. *Phys Chem Chem Phys* **17**, 7606-7618 (2015).
5. Wang, H.L., Shu, Q., Frieden, C. & Gross, M.L. Deamidation Slows Curli Amyloid-Protein Aggregation. *Biochemistry-US* **56**, 2865-2872 (2017).
6. Zhong, C. et al. Strong underwater adhesives made by self-assembling multi-protein nanofibres. *Nat. Nano.* **9**, 858-866 (2014).
7. An, B. et al. Diverse Supramolecular Nanofiber Networks Assembled by Functional Low-Complexity Domains. *Acs Nano* **11**, 6985-6995 (2017).
8. Chapman, M.R. et al. Role of Escherichia coli curli operons in directing amyloid fiber formation. *Science* **295**, 851-855 (2002).

Reviewers' comments:

Reviewer #1 (Remarks to the Author):

The authors have answered my previous comments and provided the requested control experiments. While the ThT control is convincing, there is still a small open issue related to the effect of ThT on DNA origami: The authors provide AFM images of DNA origami after incubation with 20 μ M ThT for 3 days in Fig. S19. However, there is no mention whether the samples were also exposed to light at a suitable wavelength (i.e. around 450 nm) during incubation. Exposure to light is of course essential to evaluate a possible photosensitizing effect of ThT on the DNA origami.

After this issue has been clarified, I recommend publication.

Reviewer #2 (Remarks to the Author):

The principle setup of the study, the use of CsgB-modified DNA origamis as a positional cue for curli nucleation remains innovative and of potential high interest, both for biotechnological applications as well as for addressing fundamental questions in curli assembly. Although in their revised manuscript Mao et al. provide additional data and removed several speculative points, I remain sceptical on the manuscript in several places.

1. A major shortcoming remains the lack of convincing TEM imaging demonstrating that CsgA fibrils are tethered to DNA origami. There are many studies to indicate that curli can be seen as high contrast fibers under negative stain EM. The lack of good quality TEM images showing DNA-fibril complexes is troubling. Also the nanogold TEM images shown in Response Figure X do not convince that binding of gold nanobeads is specific or is supportive of origami-curli complexes. When binding His-tagged CsgA fibrils, nanogold beads show a typical repetitive stepsize (see for example Chen et al. 2014, DOI: 10.1038/NMAT3912), which is not seen in the images. Instead, nanogold beads localization appears random.

Although the AFM images are indicative that DNA-fibril complexes may form, the many "independent fibers" and the crowded AFM images would ask for an independent observation that such complexes do form. As an alternative or in addition to TEM, the authors could show specificity with different origami designs. The authors mention origami designs with different numbers of NTA sites (1, 2 or 3) and in Figure 6, show the possibility of different modification sites (A, B, C, D). Can the authors show more representative images (wide field rather than crops to get a view of a population of origamis) of such batches demonstrating the number of fibers in contact with the origamis, and their location are accordingly?

2. A second point of attention remains over-interpretation or lack of statistical power for the quantitative data that is reported, in particular with respect to elongation rates, lengths and stagnation times of the different growth modes (Fig. 4). The data reported in Fig. 4 is said to be calculated from "at least 25 steps". How many fibers is this based on? The example kymographs shown in Figs 4a, b and c shown 4-10 "steps" for the recorded fibers, suggesting the statements are based on very few fibers (<5?) being measured. It is therefore not evident how representative these few fibers are, so that the argument regarding the different elongation rates and stagnation times seems overstated. Moreover, Table S1 reports these measurements with sub-Angstrom precision. What does that mean? Based on the molecular mass of CsgA, single subunit addition is expected to be in the order of \sim 2 nm. Please also note the units in Figure 4f to not fit with the examples shown in panels a-c, presumably this should be nm/s rather than nm/min?

3. The high concentrations of CsgA required compared to previously published studies remain troublesome. The Coomassie-stained SDS-PAGE of CsgA shows a very faint band only (what

concentration was loaded?; response Figure VII), and shows the band is smeared (in contrast to CsgB – response Figure XVI). This is indicative of a heterogeneous (i.e. FA – damaged, N- and/or O-formylated (see Zheng & Doucette 2016, doi: 10.1002/pmic.201500366)) protein and is a plausible explanation for the lower nucleation and elongation efficiency compared to previous reports as well as the GdHCL sample presented in Response Figure VIII. The authors argue GdHCL-denatured protein remained in an oligomeric or “active monomer form”. What is the evidence for this? GdHCL-purified CsgA is standardly used in many studies and has been shown to be monomeric. What is the “active monomer form” the authors are referring to? There is no published evidence such form exists, rather the contrary. The authors could perform mass spectrometry on their FA denatured sample to show protein integrity, or place a note in the methods pointing to the differences in polymerization kinetics compared to previous studies.

4. The very narrow dynamic range in which the authors claim to see the CsgB-nucleated polymerization versus spontaneous CsgA-dependent nucleation remains unexplained. Based on the concentration series in Supplementary Figure 10, the claimed specificity for CB-origami mediated nucleation is not apparent to this reviewer. Up to 3 micromolar, the individual origamis are clearly visible and there is no indication that fibers are statistically more associated with origami rather than random sites. Also, is the 5 micromolar image shown at the same magnification? This does not seem to be the case?

5. The issue of competition of CsgA-His with CsgB-His for the NTA-origami sites remains problematic. The authors use 1 nM CB-origami, with micromolar range CsgA-His, i.e. > 1000 fold molar excess. The affinity between the 6-His tag and NTA is low nanomolar, with a half-life of ~100 s (Knecht et al. 2009, doi: 10.1002/jmr.941; Kahn et al. 2006, doi:10.1021/ac060184I). This means that during a multiple hour incubation, CsgB will be effectively be replaced by CsgA at the origami-NTA sites.

Reviewer #3 (Remarks to the Author):

The authors have addressed my comments and revised manuscript is improved.

In supplementary Figure 8 for the revised manuscript it would assist the reader to better label the materials in each of the wells within the gels for SDS-PAGE and Western blot analysis and to also indicate the expected molecular weight for both purified molecules.

Dec. 9th, 2018

Re: Directing Curli Polymerization with DNA-Origami Nucleators

SUMMARY

We would first like to thank the editor and the three reviewers for their helpful and constructive inputs about our manuscript. Before getting into the full point-by-point response (below), we would like to summarize the key aspects of our revision experiments and our redrafting of the manuscript.

1. We used atomic force microscopy (AFM) and transmission electron microscopy (TEM) to re-examine and analyze the morphology of a single fibril at the vertices of DNA origami, thusly confirming that the CsgA fiber is indeed tethered to the CB-origami.
2. We have tested the integrity of formic-acid-denatured CsgA monomer via HPLC-MS analysis, supporting that formic acid (FA)-stored CsgA monomers are suitable for our research use. We have also re-run the SDS/PAGE to confirm that the short-term FA-stored sample showed single band instead of smeared band.
3. To address concerns about the replacement reaction between CsgA-His and CsgB-His to Ni-NTA, we have carried out additional QCM-D analysis and fluorescence microscopy imaging to provide more convincing evidences that CsgA-His does not replace the CsgB-His tethered to the DNA origami during fibril growth.

Regarding revisions to the text, we have made corresponding changes following the reviewers' suggestion, and have corrected several diction and typographical errors.

Finally, we deeply appreciate the work that the editor and reviewers have done on our behalf. As we trust you'll agree, we feel that both the rigor of our study and the quality of our manuscript have obviously improved as a result of following the thoughtful and helpful reviewer input during our revision process. Kindly see our detailed responses to this input in the point-by-point text that follows.

Reviewe1 (Remarks to the Author):

The authors have answered my previous comments and provided the requested control experiments. While the ThT control is convincing, there is still a small open issue related to the effect of ThT on DNA origami: The authors provide AFM images of DNA origami after incubation with 20 μ M ThT for 3 days in Fig. S19. However, there is no mention whether the samples were also exposed to light at a suitable wavelength (i.e. around 450 nm) during incubation. Exposure to light is of course essential to evaluate a possible photosensitizing effect of ThT on the DNA origami.

After this issue has been clarified, I recommend publication.

Responses from the authors:

RESPONSE: We thank the reviewer for the positive and insightful comments, which significantly helped us to improve our manuscript. The DNA origami imaged in Figure S19 (previous version) were indeed based on the samples for ThT analysis (Figure S18, previous version), in which the DNA origami were incubated with ThT over 3 days and were also exposed to the 450 nm light during the fluorescence monitoring.

Following the reviewer's comment, we have added this description in the caption of Figure S22.

Reviewer #2 (Remarks to the Author):

The principle setup of the study, the use of CsgB-modified DNA origamis as a positional cue for curli nucleation remains innovative and of potential high interest, both for biotechnological applications as well as for addressing fundamental questions in curli assembly. Although in their revised manuscript Mao et al. provide additional data and removed several speculative points, I remain sceptical on the manuscript in several places.

Responses from the authors:

We appreciate the constructive and insightful suggestions from the reviewer, which have helped greatly in improving both our understanding in this area and the quality of our manuscript. We have also attempted to address all the concerns raised by this reviewer in our revised manuscript.

1. A major shortcoming remains the lack of convincing TEM imaging demonstrating that CsgA fibrils are tethered to DNA origami. There are many studies to indicate that curli can be seen as high contrast fibers under negative stain EM. The lack of good quality TEM images showing DNA-fibril complexes is troubling. Also the nanogold TEM images shown in Response Figure X do not convince that binding of gold nanobeads is specific or is supportive of origami-curli complexes. When binding His-tagged CsgA fibrils, nanogold beads show a typical repetitive stepsize (see for example Chen et al. 2014, DOI: 10.1038/NMAT3912), which is not seen in the images. Instead, nanogold beads localization appears random.

RESPONSE: To address the concerns from the reviewer, we have carried out additional TEM imaging to provide more convincing evidence that CsgA fibers (marked in red) tethered to the CB-origami are indeed associated with CB-origami (marked in blue) (**Response Figure I**). In these TEM images, we observed bunches of biomolecular aggregates connected to the equilateral triangle shaped structures; examples of the short-fibril-decorated origami have been purposely marked in the figure, with corresponding zoom-in images provided on the right. Collectively, these TEM images illustrate that CsgA fibrils are indeed tethered to the vertices of DNA origami.

Response Figure I. TEM images showing that CsgA fibrils were tethered to the origami landmark. Scale bar 100 nm.

We also should point out that, in the paper highlighted by the reviewer (Chen et al. 2014, DOI: 10.1038/NMAT3912), CsgA fibrils produced by *E.coli* are mature amyloid fibers (comprising several laterally stacked fibrils) with larger length and thickness. However, in our designed experiments in this manuscript, we aimed to assess the relatively early stage of amyloid fibril polymerization (including nucleation and fibril growth) with DNA origami nucleators, so these fibrils finally tethered to CB-origami were actually much shorter and thinner (around 100 nm-500 nm, less than 5 nm in diameter). The fibrils with smaller diameter make themselves challenging to observe under TEM since they have less contrast feature compared to the larger mature fibers after negative staining. The presence of DNA origami make it even worse as DNA origami and fibrils may be subjected to different level of staining and thus they have different image contrast under TEM.

Following the reviewer's suggestion, we have also attempted to perform additional TEM imaging to reveal that the fibrils are tethered to DNA origami using AuNPs labeling (**Reponses Figure II**). Based on our own experience, it is much more difficult to label those fibrils with smaller diameter with Ni-NTA modified AuNPs (6 nm) than those mature fibers (>10 nm in diameter). However, in those obtained TEM images, we indeed observed that dark AuNPs adsorbed on the fibrils had a typical repetitive stepsize, the DNA origami and fibril exhibited smeared images, though the contrast of fibrils and origami compared to gold nanoparticles is very low.

Response Figure II. TEM images showing that gold nanoparticles specifically bound to CsgA fibrils tethered to the origami landmark. Scale bars: 200 nm.

Although the AFM images are indicative that DNA-fibril complexes may form, the many “independent fibers” and the crowded AFM images would ask for an independent observation that such complexes do form. As an alternative or in addition to TEM, the authors could show specificity with different origami designs. The authors mention origami designs with different numbers of NTA sites (1, 2 or 3) and in Figure 6, show the possibility of different modification sites (A, B, C, D). Can the authors show more representative images (wide field rather than crops to get a view of a population of origamis) of such batches demonstrating the number of fibers in contact with the origamis, and their location are accordingly?

RESPONSE: Following the reviewer's suggestion, we have carried out additional AFM imaging to provide more convincing evidence that CsgA fibers are tethered to the CB-origami with origami of different designs in a position-dependent manner (**Response Figure III**). These AFM images indeed revealed that the tethered positions of fibrils on DNA origami are in good agreement with our rational designs.

Response Figure III. Schematic of the applied origami structure containing a variable number of CsgB site-specifically anchored along the edge of the triangular DNA origami (left) and corresponding AFM images of the assembled structures.

2. A second point of attention remains over-interpretation or lack of statistical power for the quantitative data that is reported, in particular with respect to elongation rates, lengths and stagnation times of the different growth modes (Fig. 4). The data reported in Fig. 4 is said to be calculated from "at least 25 steps". How many fibers is this based on? The example kymographs shown in Figs 4a, b and c shown 4-10 "steps" for the recorded fibers, suggesting the statements are based on very few fibers (<5?) being measured. It is therefore not evident how representative these few fibers are, so that the argument regarding the different elongation rates and stagnation times seems overstated. Moreover, Table S1 reports these measurements with sub-Angstrom precision. What does that mean? Based on the molecular mass of CsgA, single subunit addition is expected to be in the order of ~2 nm. Please also note the units in Figure 4f to not fit with the examples shown in panels a-c, presumably this should be nm/s rather than nm/min?

RESPONSE: Thank you for raising these insightful questions and bringing these inconsistencies to our attention. We have revised our manuscript accordingly, with detailed explanations on how we made the revision listed below.

a) Following the reviewer's suggestion about analyzing the kinetics of fibril growth in various modes, including the departure, arrival, and independent modes, we monitored 8 fibers in each growth mode (compared to 4 fibrils applied in our previous manuscript) and measured their elongation distance and stagnation time during each step of fibril growth. Given that our aim here was to study the effect of CB-origami on the fibril growth in departure mode and arrival mode, we only calculated the elongation distance and stagnation time for fibril growth when fibrils were adjacent to the CB-origami (around 6-12 steps during the growth of a single fibril). Statistical analysis of these data collected for each mode indicated that the elongation rates for the departure and arrival modes are significantly higher than the rate during the individual mode of fiber growth (**Response Figure IV**). Note that the results from these recollected samples were highly consistent with the results based on our previously acquired data. We have included these data and discussion about their implications in the revised manuscript.

b) As the reviewer's point out, the original unit listed in Table S1 is wrong. We thus have revised corresponding information in the new Table S1.

c) Thanks for the reviewer pointing out the confusion about the "unit" indicated in Figure 4f. The unit indeed was supposed to be nm/s instead of nm/min for each data

point collected. However, to make the unit of Fig. 4f consistent with the unit in Fig. 4e (right) and Table SI (nm and min), we have purposely transformed the unit of Figure 4f from nm/s to nm/min. Anyway, we have made sure that each unit is properly indicated throughout the text.

Elongation Distance

Stagnation Time

Response Figure IV. Comparison of the elongation distance and stagnation time for fibril growth in the independent, departure, and arrival modes. Data for elongation distance and stagnation time were collected by counting 8 different fibers including more than 50 steps for each group, and are presented as the mean \pm s.e.m. ****** $P < 0.01$, two-tailed two-sample Student's t-test.

3. The high concentrations of CsgA required compared to previously published studies remain troublesome. The Coomassie-stained SDS-PAGE of CsgA shows a very faint band only (what concentration was loaded?; response Figure VII), and shows the band is smeared (in contrast to CsgB – response Figure XVI). This is indicative of a heterogeneous (i.e. FA – damaged, N- and/or O-formylated (see Zheng & Doucette 2016, doi: 10.1002/pmic.201500366)) protein and is a plausible explanation for the lower nucleation and elongation efficiency compared to previous reports as well as the GdHCL sample presented in Response Figure VIII. The authors argue GdHCL-denatured protein remained in an oligomeric or “active monomer form”. What is the evidence for this? GdHCL-purified CsgA is standardly used in many studies and has been shown to be monomeric. What is the “active monomer form” the authors are referring to? There is no published evidence such form exists, rather the contrary. The authors could perform mass spectrometry on their FA denatured sample to show protein integrity, or place a note in the methods pointing to the differences in polymerization kinetics compared to previous studies.

RESPONSE: We appreciate the reviewer’s insightful points.

a) The concentration of CsgA in the previous SDS/PAGE is around 0.3 mg/ml, and the smeared band in the SDS/PAGE were likely due to the use of long-term FA-stored sample (storage time > 2 weeks and the sample really suffering from different levels of deamidation), or inappropriate way to prepare the sample (For example, not fully drying the sample which might still contain small amount of formic acid). Therefore, we have performed another SDS-PAGE for CsgA samples (stored in FA for less than 2 days), and indeed found that only single band instead of smeared band appeared (Response Figure VIII).

b) Regarding the use of FA-denatured sample instead of GdHCl-stored CsgA, we realized that our previous “wrong” statements about GdHCl-stored CsgA might have caused confusions and even misunderstandings. In the revision, we first would like to clarify the reason why we chose FA-denatured sample instead GdHCl-stored CsgA in our experiment:

As the reviewer pointed out, GdHCl-stored CsgA had been used in many studies and has been shown to be monomeric. But GdHCl-stored sample often involves an additional desalting step before use (protein aggregation actually starts once GdHCl salt is removed)^{1,2}. This desalting and protein recollection requires extra some time. In addition, this approach also requires a reassessment of the collected protein

concentration, as any proteins trapped in the column will reduce the practical amount of purified proteins in solution during desalting process. We were concerned that this delayed use of CsgA monomers (even though the period is very short) would likely result in the formation of CsgA oligomers in the purified samples finally obtained.

In contrast, the extensive experimental use of formic acid (FA) and hexafluoroisopropanol (HFIP) for denaturing CsgA to keep CsgA in a monomeric state has also been quite well known³. FA-based monomer preparation protocols do not involve any desalting step. The samples can be applied for use in our experiments after re-dissolution in appropriate buffer solution. Based on these considerations, we thus chose the FA-denatured approach for our research that aims to probe DNA origami nucleators-mediated CsgA polymerization.

Following the reviewer's suggestion, to check whether FA may damage the CsgA structure, we performed HPLC-MS analysis of CsgA to evaluate protein integrity (**Response Figure V**). We first analyzed freshly purified CsgA (freshly purified via GdHCl approach without any storage time) as an internal reference. As indicated, there was only peak (around 18.5 min) in the HPLC trace (Response Figure IV a), in agreement with previous report about the typical peak for freshly purified CsgA in MS spectrum (**Response Figure V b**)². This MS spectrum thus can serve as a standard curve to evaluate if the FA-stored CsgA has been deamidated or not.

We next analyzed FA-stored CsgA samples (three days of storage, on account of the wait in the sample queue in our mass spectrometry core facility) and observed two peaks in the HPLC trace: the retention time for the first, larger peak (Integral area = 54.86) occurred at 18.3 min, while the second, much smaller peak (Integral area = 5.00) occurred around 20 min (**Response Figure V c and d**). Comparison of the spectral data indicated that the peak A stands for CsgA monomers and that of peak B is deamidated CsgA². These results thus suggest that less than 10% of the prepared CsgA monomers have been deamidated during sample preparation after 3-day storage in FA. Note that the typical storage time for monomers used in our origami experiments was much shorter (typically 12~24 hours), so this deamidation ratio should be even smaller. Therefore, deamidation, if any, should have little influence on DNA-Origami nucleator-mediated polymerization. We have noted in the methods session regarding our rationale for choosing FA instead of GdHCl for CsgA monomers storage in the revised manuscript. We have also placed a note in the methods pointing to the differences in protein storage in our manuscript compared to previous studies.

Note: the typical GdHCl approach for CsgA monomers stock^{1,2}: the CsgA monomers bound to metal affinity resin are eluted with 200 mM imidazole in 6 M GdHCl, 50

mM KPO₄ (pH 7.2), and the elution stocks are usually stored at 4 °C to avoid self-assembly. Immediately before spectroscopy or other experiments, the protein was desalted and changed to buffer of 10 or 50 mM KPO₄ (pH 7.2) using a Desalt column.

Response Figure V. a & c, HPLC traces for fresh-purified CsgA and FA-stored CsgA, respectively. b, d & e, Mass spectra for fresh-purified CsgA and FA-storage CsgA, respectively.

4. The very narrow dynamic range in which the authors claim to see the CsgB-nucleated polymerization versus spontaneous CsgA-dependent nucleation remains unexplained. Based on the concentration series in Supplementary Figure 10, the claimed specificity for CB-origami mediated nucleation is not apparent to this reviewer. Up to 3 micromolar, the individual origamis are clearly visible and there is no indication that fibers are statistically more associated with origami rather than random sites. Also, is the 5 micromolar image shown at the same magnification? This does not seem to be the case?

RESPONSE: We appreciate the reviewer's insightful points. Indeed, the 5 micromolar image is shown at the same magnification.

To explain the fact that independent fibril polymerization became dominant when CsgA concentration was above 3 μM , we constructed a kinetic model based on our ThT data, through which a curve representative of the function of aggregate numbers produced over time is established. The aggregate number represents the cumulative number of aggregates based on all nucleation events including the independent nucleation events and DNA origami nucleator-mediated nucleation events. Following the paper of Knowles *et al.* (2009)^{4,5}, we calculated the total number of all aggregates at various concentrations in the presence of CB-origami (Total aggregate number = $t * K_{\text{aggregates}} = t * (K_{\text{primary}} + K_{\text{secondary}}) = t * ([\text{CsgA}] * K_n + [\text{CsgA}] * [\text{Fiber}] * K_2)$, in which K represents the formation rate of aggregates and t represents the aggregation time) (**Response Figure VI**).

As indicated from Figure VI, around 0.91 nM of aggregates can be produced at 24 hour (the time that we collected the sample for AFM imaging) when 3 μM CsgA is applied, while only 0.38 and 0.18 nM of aggregates are obtained in the system for 2 and 1 μM CsgA, respectively. As the concentration for the added CB-origami in the system is 0.2 nM, and if we assume that all the CB-origami serve as effective nucleators in the systems, then aggregates produced based on independent nucleation events for 3 μM CsgA (0.38- 0.2 = 0.71 nM) is significant higher compared to 2 μM CsgA (0.38 - 0.2 = 0.18 nM) and 1 μM CsgA (0). The aggregates produced based on independent nucleation events for 5 μM CsgA is even larger (2.1 nM). Taken together, the results suggest that the number of CsgA aggregates based on independent nucleation events is significantly higher than that of DNA origami nucleator-mediated aggregates when the concentration of CsgA is over 3.0 μM (and note that the increase in the number of independent aggregates is not proportional to the increase of CsgA concentration), thus explaining the trend that a significantly increased extent of

self-assembled fibrils were indeed found on the mica surface under AFM imaging as CsgA concentration increased.

Response Figure VI. A kinetic model (number of aggregates/ time) for CsgA polymerization with 1.0, 2.0, and .0 μM CsgA concentrations in the presence of CB-origami. The blue dot indicates the concentration of CB-origami applied in our experiment.

5. The issue of competition of CsgA-His with CsgB-His for the NTA-origami sites remains problematic. The authors use 1 nM CB-origami, with micromolar range CsgA-His, i.e. > 1000 fold molar excess. The affinity between the 6-His tag and NTA is low nanomolar, with a half-life of ~100 s (Knecht et al. 2009, doi: 10.1002/jmr.941; Kahn et al. 2006, doi:10.1021/ac060184l). This means that during a multiple hour incubation, CsgB will be effectively replaced by CsgA at the origami-NTA sites.

Response: We appreciate the reviewer's insightful points. To determine whether CsgA-His would replace CsgB-His that are initially tethered to the Ni-NTA sites in the DNA origami, we applied quartz crystal microbalance (QCM) analysis. In the QCM curve, the frequency change of a quartz crystal resonator shows the mass change, which is directly correlated to the replacement reaction.

Given that CsgA and CsgB are of almost the same size (~14 kDa) and are thus not easily differentiated via QCM even if the replacement reactions occur, we therefore designed and conducted the replacement assay by applying high-concentration of GFP-His to replace CsgB-His proteins that were initially bound with Ni-NTA-decorated Au chip (**Response Figure VII a and b**). GFP-His has a molecular weight around 28 kDa (substantially larger than CsgA-His (14 kDa), and thus this replacement reaction, if occurs, should be more amenable for observation based on QCM measurements.

Specifically, we initially incubated CsgB-His with Ni-NTA-modified gold chip for 3 hours to ensure that Ni-NTA sites on the surface were fully occupied with CsgB-His monomers. After washing away the loosely bound CsgB-His on the surfaces with copious amount of buffer, we applied constant flow of GFP-His solution in the system to carry out the replacement reaction by incubating the CsgB-His tethered Ni-NTA modified gold chip in the presence of 3.0 μM GFP-His solution (this concentration was purposely selected to match the concentration that we applied in our experiment in Figure 2-4.) We then reapplied large amount of buffer (with constant flow rate) to wash away any loosely bound GFP-His protein. As indicated, GFP-His proteins in solution could first adsorb on the quartz's surface (indicated by the decrease in frequency), but almost all the adsorbed GFP-His proteins could be washed away with buffer (as indicated by the frequency fully recovered to the original position before incubation with GFP-His protein solution). These results clearly indicated that 3.0 μM GFP-His could not replace CsgB-His that was initially bound with Ni-NTA decorated Au surface.

Given that GFP protein has fluorescent feature, we further applied fluorescence microscopic imaging to directly observe if GFP-His can replace CsgB-His that was

pre-bound to Ni-NTA decorated surfaces (Response Figure VII c). To such ends, we chemically modified the gold surface with HS-NTA via the formation of thiol-Au bond. The Ni-NTA modified gold surface was then incubated with CsgB-His in a region-defined manner, in which only half of the gold surfaces was exposed to excessive amount of CsgB-His proteins. We then applied excessive amount of 3.0 μ M GFP-His solution to cover both areas and the substrate with covered solution was incubated at ambient condition for another 6 hours. The CsgB-incubated area exhibited no GFP fluorescence feature while the other area was fully covered with GFP fluorescence, indicating that the GFP-His did not replace CsgB-His from the Ni-NTA decorated surfaces.

The above experimental results, along with the nucleation experimental results in Figure 2b, in which CB-origami exhibited significantly higher nucleation efficiency than that of CA-origami, clearly implied that CsgA-His did not replace CsgB in the CB-origami during CsgA polymerization process.

In summary, we thus conclude that the replacement of CsgA-His with CsgB-His from Ni-NTA-decorated DNA origami is a pretty slow process that wouldn't affect our experiment design for applying CsgB-His tethered origami for directing CsgA-His polymerization.

c) Fluorescence microscopy imaging

Ni-NTA Au surface

Response Figure VII. a, Cartoon showing the application of a Ni-NTA decorated Au sensor chip to probe the replacement of CsgB-His (originally tethered to the Ni-NTA sites) with a high concentration of GFP-His based on QCM technique. b, Real-time frequency response of the Au chip sensor upon exposure to different protein solutions or washing buffer under constant flow rate showing different adsorption (binding) and de-adsorption (de-binding) events. The arrows refer to the time points at which CsgB-His protein solution, (washing) buffer or GFP-His solution was flowed in. c, Fluorescent imaging used to determine if GFP-His could replace the CsgB-His initially tethered to the Ni-NTA decorated Au Chip. The left half of the chip was pre-incubated with CsgB-His solution, and the right half of the chip was the control Ni-NTA decorated Au chip. The whole chip was subjected to incubation in 3.0 μM GFP-His solution before imaging.

Reviewer #3 (Remarks to the Author):

The authors have addressed my comments and revised manuscript is improved.

In supplementary Figure 8 for the revised manuscript it would assist the reader to better label the materials in each of the wells within the gels for SDS-PAGE and Western blot analysis and to also indicate the expected molecular weight for both purified molecules.

Responses from the authors:

Thank you for the positive and encouraging comments, for the constructive suggestions, and for supporting the publication of our work.

Following the reviewer's suggestion, we have labelled the materials in the SDS-PAGE and Western blot analysis in our revised manuscript, and we have included these data in the revised manuscript (**Response Figure VIII**).

Response Figure VIII. Purification and biological assays of purified CsgA and CsgB proteins. (a) & (b)Coomassie-stained SDS-PAGE and western blots with anti-His antibodies confirming the expression of the proteins: CsgA (a) and CsgB (b); (c) ThT assay revealing the kinetics of amyloid formation for CsgA. Kinetic model (aggregate numbers/time) for CsgA polymerization at a 5.0 μ M CsgA concentration.

References:

1. Q. Shu, S. L. Crick, J. S. Pinkner, B. Ford, S. J. Hultgren and C. Frieden, ***Proceedings of the National Academy of Sciences of the United States of America***, 2012, 109, 6502-6507.
2. H. Wang, Q. Shu, C. Frieden and M. L. Gross, ***Biochemistry***, 2017, 56, 2865-2872.
3. M. R. Chapman, L. S. Robinson, J. S. Pinkner, R. Roth, J. Heuser, M. Hammar, S. Normark and S. J. Hultgren, ***Science***, 2002, 295, 851-855.
4. T. P. J. Knowles, C. A. Waudby, G. L. Devlin, S. I. A. Cohen, A. Aguzzi, M. Vendruscolo, E. M. Terentjev, M. E. Welland and C. M. Dobson, ***Science***, 2009, 326, 1533-1537.
5. S. I. A. Cohen, S. Linse, L. M. Luheshi, E. Hellstrand, D. A. White, L. Rajah, D. E. Otzen, M. Vendruscolo, C. M. Dobson and T. P. J. Knowles, ***Proceedings Of the National Academy Of Sciences Of the United States Of America***, 2013, 110, 9758-9763.

Reviewers' comments:

Reviewer #1 (Remarks to the Author):

The authors have addressed all my previous comments. I recommend publication of the manuscript in its present form.

Reviewer #2 (Remarks to the Author):

The authors have made a substantial effort at clarifying my concerns raised in the 2nd review. In several places this does take away previous points of concern, in others, I'm afraid I remain doubtful of the interpretation presented by the authors. I appreciate the experimental difficulty in providing unambiguous proof to the arguments made in the paper and the sincere efforts of the authors to do so. Nevertheless, I remain sceptical based on the current data. At the same time, I realise this view differs to that of the other reviewers, so that I would propose that the manuscript is reviewed by a fourth referee that has a strong expertise in AFM imaging.

1. Additional TEM and AFM images in support to specific tethering of curli fibers to CsgB-modified DNA origamis.

a. Response Figure I. The TEM images do show clear curli. Although the negative stain the images remain ambiguous whether these are associated to DNA origamis, the co-staining with NTA-gold in Response Figure II appears to suggest they might be. In the second (lower panel), the repeat structure of the curli fibers is now also apparent.

b. AFM – Response Figure III. The AFM images on the type B origamis are convincing that curli fibers have a position-dependent association with the DNA origamis. For the other designs (AB, AC, ADC) however, this is not evident. Are the thin threads shown in between the origamis curli? They appear different in morphology (width, height and curvature) to those seen in the upper panel and elsewhere in the manuscript. Could this be DNA from non- or partially assembled origamis? Also in these panels, although the circled areas show threads that contact the origamis in the predicted positions, there are at least as many contacts in the field of view that are in disagreement with the position of the NTA modification.

It would be better to remove these panels from the Supplementary information and only show the type B origamis as an alternative to the apically labelled origamis in most of the main figures. The same comment applies to main Figure 6, panel b, where the threads visible differ substantially in morphology to canonical curli shown elsewhere in the manuscript and in other studies.

2. The authors now provide an exact description and a higher number of data points for the statistical analyses of fiber elongation kinetics in the different proposed growth regimes.

3. Quality control of the CsgA samples

a. In their study, the authors use CsgA preps originating from FA-dissolved curli fibers. This is indeed a strategy previously used in other studies, although it has a substantial drawback due to the risk of acid and oxidation-induced damage to CsgA preps. This is a particular concerns since the single fiber polymerization and nucleation kinetics and the required CsgA concentrations described here are orders of magnitude different to those reported previously. In the revised manuscript, the authors now provide HPLC-MS data to compare integrity of the CsgA preps. It is not clear what the HPLC traces and MS spectra correspond to however. The A peak has a m/z of ~ 600 ($z=2$) which is compatible with a smaller organic such as a peptide, but not the full length CsgA protein. These data do not support the claim that the FA dissolved material is fully active. Based on the discrepancies in polymerization kinetics with previous studies this is indeed doubtful and rather suggest that only a minor fraction of the protein preps is active. Even so, this would have little influence on the qualitative aspects of the manuscript. However, it does have significant influence on the kinetic analyses in the manuscript. Assuming that the same or very similarly produced and aged CsgA preps were used for the different kinetic regimes, this would be OK for relative comparison within the manuscript, but the authors would do well to place a note that the reduced polymerization kinetics and high concentrations may be due to partially damaged protein

samples as a result of HA treatment. Supplementary Fig 10 needs to be clarified or removed from the manuscript. The data is incompatible with representing “freshly purified or FA-stored CsgA”.

4. CsgB-origami dependent nucleation of CsgA and Supplementary Fig. 11

a. This image remains problematic. In the 2 micromolar panel, I see just one, maybe two curly fibers that could be tethered to a DNA origami, compared to at least 16 fibers that have no contact with an origami apex. At 3 micromolar, there is again no indication that fibers are statistically more likely to be tethered to a DNA apex. Based on these images, I see no evidence to the claim that CsgB-labelled DNA origamis specifically nucleate CsgA fibers. The presented kinetic model and ThT experiments cannot provide evidence to CsgB-origami-based nucleation versus nucleation by CsgB dissociated from origamis.

5. Equilibration of CsgB-His – Ni-NTA contacts

a. In response to the question whether the 1000-fold excess in CsgA-his could have competed off CsgB-his at the origami-NTA sites the authors now provide a quartz crystal microbalance analysis of CsgB interaction to a Ni-NTA-decorated Au chip and its possible displacement by GFP-his. However, this analysis is flawed since the Au chip has a multivalent display of Ni-NTA sites and CsgB is known to rapidly polymerize. It is quite likely that the authors were not measuring the equilibration of monovalent 6-His – Ni-NTA interactions, but were rather competing a monovalent (GFP-his) affinity-driven ligand with a multivalent (CsgB-His polymer), avidity-driven interaction. The monovalent 6-His – Ni-NTA dissociation/association rates and affinities have been described multiple times. The affinity is reported by independent studies as low nanomolar, with a half-life of ~100 s (Knecht et al. 2009, doi:10.1002/jmr.941; Kahn et al. 2006, doi:10.1021/ac060184l). The observation that GFP-his does not displace CsgB-his from a Ni-NTA Au chip is indicative that CsgB has indeed polymerized and has a much higher avidity than the monovalent 6-his – Ni-NTA affinity. At the DNA origamis, the interaction is monovalent however, such that the concern on equilibration of CsgA-his and CsgB-his remains.

This experiment shown in Supplementary 20 therefore does not “clearly imply that CsgA-His did not replace CsgB in the CB-origami during CsgA polymerization”. Unless unambiguous proof is provided, for example by specific labelling, the authors would be more prudent to include a note in the main text that such equilibration reaction is a possibility that cannot currently be excluded.

Feb. 7th, 2018

Re: Directing Curli Polymerization with DNA-Origami Nucleators [NCOMMS-18-12416B]

SUMMARY

We would first like to thank the editor and the reviewers for their helpful and constructive inputs about our manuscript. Before getting into the full point-by-point response (below), we would like to summarize the key aspects of our revision experiments and our redrafting of the manuscript.

1. Following reviewer's suggestion, we added a few sentences stating the advantages and disadvantages of applying formic acid (FA) to denature CsgA instead of 8M guanidine hydrochloride (GdHCl) in our experimental section.
2. Following reviewer's suggestion, we added statements about the non-ideal decoration of CB-origami using non-covalent reactions between Ni-NTA and Histidine in our design, as well as comments about the limitations and difficulties in evaluating the replacement reaction between CB-origami and CsgA-his.

Finally, we deeply appreciate the work that the editor and reviewers have done on our behalf. As we trust you'll agree, we feel that both the rigor of our study and the quality of our manuscript have obviously improved as a result of following the thoughtful and helpful reviewer input during our revision process. Kindly see our detailed responses to this input in the point-by-point text that follows.

Editorial note: Reviewer #2 proposed to have an expert in Atomic Force Microscopy check over their comments. Following this advice, the editor asked Reviewer #1, who kindly agreed to respond to the comments.

Q1 :

1. Additional TEM and AFM images in support to specific tethering of curli fibers to CsgB-modified DNA origamis.

a. Response Figure I. The TEM images do show clear curli. Although the negative stain the images remain ambiguous whether these are associated to DNA origamis, the co-staining with NTA-gold in Response Figure II appears to suggest they might be. In the second (lower panel), the repeat structure of the curli fibers is now also apparent.

[Reviewer 1]

I find it quite difficult to identify the fibrils and the origami in these TEM images, but the data seems to further support the other results obtained by alternative/complementary techniques. So I'm OK with that.

Response from the authors:

We very much appreciate reviewer 1's support for our supplemental TEM and AFM images.

Q2:

b. AFM – Response Figure III. The AFM images on the type B origamis are convincing that curli fibers have a position-dependent association with the DNA origamis. For the other designs (AB, AC, ADC) however, this is not evident. Are the thin threads shown in between the origamis curli? They appear different in morphology (width, height and curvature) to those seen in the upper panel and elsewhere in the manuscript. Could this be DNA from non- or partially assembled origamis?

[Reviewer 1]

No. Residual non-assembled DNA comes in two flavors:

- unfolded scaffold: this has a peculiar morphology due to pronounced secondary structure and can be clearly identified in AFM. Furthermore, under standard assembly conditions, it is virtually impossible to obtain completely unfolded scaffold.
- staples: these would be short single strands which cannot be resolved with “normal” AFM, as the authors used here.

From these AFM images, I think it's pretty clear that these “threads” are indeed protein filaments. Why they have different morphologies, however, I can only guess. From my experience with pathogenic amyloid, such differences may result from variations in sample handling. I don't know about curli, though. Then again, in the Methods part, the authors mention that AFM was performed in air or liquid. Hydration and dehydration of biomolecules can cause drastic changes in morphology and especially in height. Unfortunately, the authors don't mention how the individual images in Response Figure III were recorded, but from the varying image quality, I assume that what we see here is a mix of liquid and dry mode images. So this could be an explanation as well.

Response from the authors:

Firstly, we would like to thank Reviewer 1 for supporting us in our view that that the “threads” shown in the AFM images in Response Figure III are protein filaments. We also thank Reviewer 1 for helping to clarify things by detailing potential influences which may have led to the morphological differences of fibril threads as shown in AFM images. Indeed, these images in Response Figure III were AFM images recorded in air mode, and dehydration in dry condition may occur in amyloid fibrils. Therefore, the morphological differences of fibril threads may come from variations in sample handling (for example, nanofiber shrinkage arising from a dehydration effect with N₂ blowing or the air mode used in imaging), AFM tip artifacts (for example, the actual AFM tip radius may vary from batch to batch), as suggested by Reviewer 1.

In this revision, we will clearly specify the information about the imaging modes in the figure captions and will incorporate the aforementioned information and considerations in the Methods section as well in the main text of the revised manuscript.

Q3:

Also in these panels, although the circled areas show threads that contact the origamis in the predicted positions, there are at least as many contacts in the field of view that are in disagreement with the position of the NTA modification.

Reviewer 1

I agree, there are also fibrils that do not terminate at the expected positions of the DNA origami. However, some of the images do reveal a rather strong branching of the fibrils, so it is not surprising that such branching and (possibly, at least for pathogenic amyloids) secondary nucleation events trigger the formation of new fibrils in solution that grow without direct contact to DNA origami seeds. Also it seems that spontaneous aggregation happens in bulk, independent of the DNA origami seeds (see below). But either way, I find the AFM images (in particular the new ones in Response Figure III) very convincing and am completely certain that DNA origami-bound initiators can stimulate fibril growth from the DNA origami substrate. This is clearly demonstrated. This process may not be stoichiometric, and there may other stuff be going on in bulk solution, but that does not change the fact that the approach in general works.

Response from the authors:

We thank reviewer 1's help in clarifying that secondary nucleation might trigger the formation of new fibril growth in solution without direct contact to DNA origami seeds. We particularly appreciate the reviewer's strong support that our approach generally works based on the AFM images provided.

Q4:

It would be better to remove these panels from the Supplementary information and only show the type B origamis as an alternative to the apically labelled origamis in most of the main figures. The same comment applies to main Figure 6, panel b, where the threads visible differ substantially in morphology to canonical curli shown elsewhere in the manuscript and in other studies.

[Reviewer 1]

This particular image (Fig. 6b) does show quite some tip artefacts which I would blame for the different morphologies. Replacing a bad image by a better one is always a good idea, but I wouldn't say that it's absolutely necessary.

Response from the authors:

We indeed believe that the AFM tip artifacts and the dehydration may result in different morphologies of fibril threads shown in figure 6 and supplemental Fig. 27. To reveal a clearer view of the fibril thread tethered to DNA origami, we flattened the AFM image in Figure 6b. In addition, we have also replaced the old AFM image of the type B origamis with a better AFM image.

Following the reviewer's suggestion, we have also added some explanation about the artifacts (main text, Page 11) and statement on the difficulty in collecting these AFM images (Supplemental figure 27, figure caption). The revisions are listed below as a reference:

Notably, all the AFM images collected in the Figure 6 were carried out in air mode, the AFM imaging in dry condition and possible dehydration during sample preparation, unfortunately, have led to varied morphologies of fibril threads shown in AFM images. Nevertheless, these AFM images indeed demonstrated various fibril/DNA origamis structures could be constructed other than the apically labelled origamis by taking advantage of the nucleation role of CsgB protein (Fig. 6a, b and Supplemental Fig. 27 and 28).

Q5:

1. The authors now provide an exact description and a higher number of data points for the statistical analyses of fiber elongation kinetics in the different proposed growth regimes.

a. Quality control of the CsgA samples. In their study, the authors use CsgA preps originating from FA-dissolved curli fibers. This is indeed a strategy previously used in other studies, although it has a substantial drawback due to the risk of acid and oxidation-induced damage to CsgA preps. This is a particular concern since the single fiber polymerization and nucleation kinetics and the required CsgA concentrations described here are orders of magnitude different to those reported previously. In the revised manuscript, the authors now provide HPLC-MS data to compare integrity of the CsgA preps. It is not clear what the HPLC traces and MS spectra correspond to however. The A peak has a m/z of ~600 ($z=2$) which is compatible with a smaller organic such as a peptide, but not the full length CsgA protein. These data do not support the claim that the FA dissolved material is fully active. Based on the discrepancies in polymerization kinetics with previous studies this is indeed doubtful and rather suggest that only a minor fraction of the protein preps is active. Even so, this would have little influence on the qualitative aspects of the manuscript. However, it does have significant influence on the kinetic analyses in the manuscript. Assuming that the same or very similarly produced and aged CsgA preps were used for the different kinetic regimes, this would be OK for relative comparison within the manuscript, but the authors would do well to place a note that the reduced polymerization kinetics and high concentrations may be due to partially damaged protein samples as a result of HA treatment. Supplementary Fig 10 needs to be clarified or removed from the manuscript. The data is incompatible with representing “freshly purified or FA-stored CsgA”.

[Reviewer 1]

Well, I can't say much about the prep, but adding a few sentences stating that the prep might not be ideal for known reasons as suggested by the reviewer seems like a reasonable and straightforward thing to do. And as the reviewer mentioned, as long as they used the same protocol/prep/stock throughout the paper, they still can draw solid conclusions concerning relative effects (e.g. origami vs. no origami), which is the main point of the paper. So I think that's no big concern.

Response from the authors:

Following the reviewer's suggestion, we add a few sentences stating the advantages and disadvantages of applying formic acid to denature CsgA instead of 8M GdHCl in our experimental section. The statements were listed below as a reference:

“In particular, we applied formic acid (FA) to dissolve and store CsgA monomers instead of the widely used guanidine hydrochloride (GdHCl) in our study, as, unlike GdHCl-stored CsgA, the FA-stored CsgA does not require an additional time-consuming desalting process. (**main text, page 5**)

“These results suggest that most CsgA stored in formic acid remain integrity as fresh purified CsgA, indicating that deamination process has little influence on the integrity of CsgA. (**Supplemental Fig 10**)

“In our study, we applied an approach of CsgA monomer storage in formic acid, which might reduce polymerization kinetics and require the relatively higher concentrations of CsgA for

CB-origami-triggered CsgA polymerization. However, as we indeed applied the same protocol/prep/stock methods for CsgA monomers throughout the whole study, therefore the major conclusions concerning the relative effects (e.g. origami vs. no origami) were still valid in our paper.” (**main text, page 12**).

Q6

2. CsgB-origami dependent nucleation of CsgA and Supplementary Fig. 11

a. This image remains problematic. In the 2 micromolar panel, I see just one, maybe two curly fibers that could be tethered to a DNA origami, compared to at least 16 fibers that have no contact with an origami apex. At 3 micromolar, there is again no indication that fibers are statistically more likely to be tethered to a DNA apex. Based on these images, I see no evidence to the claim that CsgB-labelled DNA origamis specifically nucleate CsgA fibers. The presented kinetic model and ThT experiments cannot provide evidence to CsgB-origami- based nucleation versus nucleation by CsgB dissociated from origamis.

Again, there is definitely spontaneous aggregation occurring in bulk, independent of the DNA origami. This is clearly stated by the authors, but does not mean that the CB-origami have no effect at all. It's just not a dominant effect under most of the conditions shown here. However, the other AFM images recorded under optimized conditions clearly show fibrils terminating at the CsgB modification site of the DNA origami. Whether the spontaneous aggregation in bulk is triggered by dissociated CsgB, I don't know. But even if that's the case, would that mean that the general approach doesn't work? I'd say it just means that tethering the CsgB to the origami needs to be optimized.

4. Equilibration of CsgB-His – Ni-NTA contacts

a. In response to the question whether the 1000-fold excess in CsgA-his could have competed off CsgB-his at the origami-NTA sites the authors now provide a quartz crystal microbalance analysis of CsgB interaction to a Ni-NTA-decorated Au chip and its possible displacement by GFP-his. However, this analysis is flawed since the Au chip has a multivalent display of Ni-NTA sites and CsgB is known to rapidly polymerize. It is quite likely that the authors were not measuring the equilibration of monovalent 6-His – Ni-NTA interactions, but were rather competing a monovalent (GFP-his) affinity-driven ligand with a multivalent (CsgB-His polymer), avidity-driven interaction. The monovalent 6-His – Ni-NTA dissociation/association rates and affinities have been described multiple times. The affinity is reported by independent studies as low nanomolar, with a half-life of ~100 s (Knecht et al. 2009, doi:10.1002/jmr.941; Kahn et al. 2006, doi:10.1021/ac0601841). The observation that GFP-his does not displace CsgB-his from a Ni-NTA Au chip is indicative that CsgB has indeed polymerized and has a much higher avidity than the monovalent 6-his – Ni-NTA affinity. At the DNA origamis, the interaction is monovalent however, such that the concern on equilibration of CsgA-his and CsgB-his remains. This experiment shown in Supplementary 20 therefore does not “clearly imply that CsgA-His did not replace CsgB in the CB-origami during CsgA polymerization”. Unless unambiguous proof is provided, for example by specific labelling, the authors would be more prudent to include a note in the main text that such equilibration reaction is a possibility that cannot currently be excluded.

[Reviewer 1]

The reviewer may have a point here, although I have never worked with His-tags myself. But again, pointing out the difficulties of properly controlling the system is a straightforward thing to do, and I'm always in favor of

clearly stating limitations and difficulties. So I think that's a decent solution to this issue.

Response from the authors:

We thank Reviewer 1's constructive suggestions. We particularly thank the reviewer for his understanding of the challenges in assessing the replacement equilibrium reaction between CB-origami and CsgA and for offering us suggestion that we state the limitations and difficulties in evaluating such reactions with our designed QCM-D experiments.

Admittedly, our QCM-D experiments were not ideal given the possibility that CsgB-his proteins tend to form nanofibers on the Au chip (even though the incubation time is short) and in light of the multivalent display of Ni-NTA sites on the chips. Unfortunately, we have not been able to come up with any better characterization than QCM-D experiments to demonstrate if such a replacement equilibrium reaction does dominate or not in the system. The similarities in the molecular weight and structure between CsgB and CsgA make it even more challenging to distinguish them from each other using routine characterization tools. We would consider the decoration of CsgB with certain fluore (e.g. fluorescent dyes) without significantly changing the structure and function of CB-origami, but such experiments would take more time and may not yield meaningful conclusions because they would not alter our current major aforementioned conclusion that DNA origami-bound initiators can stimulate fibril growth from the DNA origami substrate.

Given that we don't have sufficient or unambiguous evidence to rule out the possibility of this equilibration reaction, we have added statements in our manuscript about the non-ideal decoration of CB-origami using non-covalent reactions between Ni-NTA and Histidine in our design, as well as comments about the limitations and difficulties in evaluating the replacement reaction between CB-origami and CsgA-his. These statements were listed here as a reference: However, our QCM-D experiments were not ideal given the possibility that CsgB-his proteins tend to form nanofibers on the Au chip (even though the incubation time is short) and in light of the multivalent display of Ni-NTA sites on the chips. It is indeed very challenging to demonstrate if such a replacement equilibrium reaction does dominate or not in the system, particularly considering that the similarities in the molecular weight and structure between CsgB and CsgA. (**Supplemental Fig. 20**)

Last but not least, we should also point out that the nucleation role of CB-origami indeed takes effect even if a certain degree of replacement reaction occurs in our system. As shown in Figure 3b, the addition of CsgB monomers usually shortens the lag time (faster nucleation of CsgA) and speeds up subsequent aggregation of CsgA to reach the plateau compared to the addition of the CB-origami in the ThT dynamic curves. This slower nucleation in CB-origami than CsgB monomers may arise from the hindrance effect of DNA origami. Because the actual CsgB concentration was the same in the two cases, if the replacement reaction between CsgA and CB-origami dominates, a similar ThT dynamic curve would be obtained. These results therefore suggest a quite different nucleation and polymerization mechanism in the two cases, indirectly ruling out the possibility that CB-origami were mainly replaced by CsgA-his in solution. Anyway, these ThT results (Fig 3b), along with nucleation efficiency comparison (Fig. 2b) and real-time AFM imaging (Figure 4), can indeed lead to solid conclusions concerning relative effects (e.g.

origami vs. no origami) in triggering CsgA polymerization.

We have also added corresponding statements in the manuscript (**main text, Page 12, Paragraph 2**), which are listed below as a reference “In addition, the construction of CB-origami through decoration of CsgB to DNA origami via Ni-NTA metal coordination chemistry was not ideal in our study given the potential replacement reaction between CB-origami and CsgA in solution. However, the ThT results in Figure 3b, along with nucleation efficiency comparison (Fig. 2b) and real-time AFM imaging (Figure 4), could indeed lead to solid conclusions concerning relative effects (e.g. origami vs. no origami) in triggering CsgA polymerization. We therefore conclude that the designed DNA origami nucleators, CB-origami, can stimulate fibril growth from the DNA origami substrate.”